# Communication-Efficient Topologies for Decentralized Learning with $\mathcal{O}(1)$ Consensus Rate

**Zhuoqing Song**[1]*, **Weijian Li**[2]*, **Kexin Jin**[3]*, **Lei Shi**[1,7], **Ming Yan**[4,5], **Wotao Yin**[2], **Kun Yuan**[2,6]

[1]Fudan University  [2]Alibaba DAMO Academy  [3]Princeton University
[4]The Chinese University of Hong Kong, Shenzhen  [5]Michigan State University
[6]Peking University  [7]Shanghai Artificial Intelligence Laboratory

zqsong19@fudan.edu.cn, weijian.li@alibaba-inc.com, kexinj@math.princeton.edu
yanming@cuhk.edu.cn, leishi@fudan.edu.cn
wotao.yin@alibaba-inc.com, kun.yuan@alibaba-inc.com

## Abstract

Decentralized optimization is an emerging paradigm in distributed learning in which agents achieve network-wide solutions by peer-to-peer communication without the central server. Since communication tends to be slower than computation, when each agent communicates with only a few neighboring agents per iteration, they can complete iterations faster than with more agents or a central server. However, the total number of iterations to reach a network-wide solution is affected by the speed at which the agents' information is "mixed" by communication. We found that popular communication topologies either have large maximum degrees (such as stars and complete graphs) or are ineffective at mixing information (such as rings and grids). To address this problem, we propose a new family of topologies, EquiTopo, which has an (almost) constant degree and a network-size-independent consensus rate that is used to measure the mixing efficiency.

In the proposed family, EquiStatic has a degree of $\Theta(\ln(n))$, where $n$ is the network size, and a series of time-dependent one-peer topologies, EquiDyn, has a constant degree of 1. We generate EquiDyn through a certain random sampling procedure. Both of them achieve an $n$-independent consensus rate. We apply them to decentralized SGD and decentralized gradient tracking and obtain faster communication and better convergence, theoretically and empirically. Our code is implemented through BlueFog and available at https://github.com/kexinjinnn/EquiTopo.

## 1  Introduction

Modern optimization and machine learning typically involve tremendous data samples and model parameters. The scale of these problems calls for efficient distributed algorithms across multiple computing nodes. Traditional distributed approaches usually follow a centralized setup, where each node needs to communicate with a (virtually) central server. This communication pattern incurs significant communication overheads and long latency.

Decentralized learning is an emerging paradigm to save communications in large-scale optimization and learning. In decentralized learning, all computing nodes are connected with some network topology (e.g., ring, grid, hypercube, etc.) in which each node averages/communicates locally with its immediate neighbors. This decentralized setup allows each node to communicate with fewer neighbors and hence has a much lower overhead in per-iteration communication. However, local averaging is less effective in "mixing" information, making decentralized algorithms converge slower

---

*Equal Contribution. Corresponding Author: Kun Yuan

Table 1: Comparison between different commonly-used topologies. "Static Exp.": static exponential graph; "O.-P. Exp.": one-peer exponential graph; "E.-R. Rand": Erdos-Renyi random graph $G(n, p)$ with probability $p = (1 + a) \ln(n)/n$ for some $a > 0$; "Geo. Rand": geometric random graph $G(n, r)$ with radius $r^2 = (1 + a) \ln(n)/n$ for some $a > 0$. Undirected graphs can admit symmetric gossip matrices. If some graph has a dynamic pattern, its associated gossip matrix will vary at each iteration.

| Topology | Connection | Pattern | Degree | Consensus Rate | size $n$ |
|---|---|---|---|---|---|
| Ring [22] | undirect. | static | $\Theta(1)$ | $1 - \Theta(1/n^2)$ | arbitrary |
| Grid [22] | undirect. | static | $\Theta(1)$ | $1 - \Theta(1/(n \ln(n)))$ | arbitrary |
| Torus [22] | undirect. | static | $\Theta(1)$ | $1 - \Theta(1/n)$ | arbitrary |
| Hypercube [31] | undirect. | static | $\Theta(\ln(n))$ | $1 - \Theta(1/\ln(n))$ | power of 2 |
| Static Exp.[37] | directed | static | $\Theta(\ln(n))$ | $1 - \Theta(1/\ln(n))$ | arbitrary |
| O.-P. Exp.[37] | directed | dynamic | 1 | finite-time conv.[†] | power of 2 |
| E.-R. Rand [22] | undirect. | static | $\Theta(\ln(n))$[◇] | $\Theta(1)$ | arbitrary |
| Geo. Rand [5] | undirect. | static | $\Theta(\ln(n))$ | $1 - \Theta(\ln(n)/n)$ | arbitrary |
| D-EquiStatic | directed | static | $\Theta(\ln(n))$ | $\rho \in (0, 1)$[‡] | arbitrary |
| U-EquiStatic | undirect. | static | $\Theta(\ln(n))$ | $\rho \in (0, 1)$[‡] | arbitrary |
| OD-EquiDyn | directed | dynamic | 1 | $\sqrt{(1 + \rho)/2}$ | arbitrary |
| OU-EquiDyn | undirect. | dynamic | 1 | $\sqrt{(2 + \rho)/3}$ | arbitrary |

[†] One-peer exponential graph has finite-time exact convergence only when $n$ is the power of 2.

[◇] $\Theta(\ln(n))$ is the averaged degree; its maximum degree can be $O(n)$ with a non-zero probability.

[‡] Constant $\rho = \Theta(1)$ is independent of network-size $n$.

than their centralized counterparts. Therefore, seeking a balance between communication efficiency and convergence rate in decentralized learning is critical.

The network topology (or graph) determines decentralized algorithms' per-iteration communication and convergence rate. The maximum graph degree controls the communication cost, whereas the connectivity influences the convergence rate. Intuitively speaking, a densely-connected topology enables decentralized methods to converge faster but results in less efficient communication since each node needs to average with more neighbors. Selecting an appropriate network topology is key to achieving light communication and fast convergence in decentralized learning.

## 1.1 Prior arts in topology selections

**Gossip matrix and consensus rate.** Given a connected network of size $n$ and its associated doubly-stochastic gossip matrix $\boldsymbol{W} \in \mathbb{R}^{n \times n}$ (see the definition in § 2), its consensus rate $\beta$ determines how effective the gossip operation $\boldsymbol{W}\boldsymbol{x}$ is to mix information (see more explanations in § 2). It is a long-standing topic in decentralized learning to seek topologies with both a small maximum degree and a fast consensus rate (i.e., a small $\beta$ as close to 0 as possible).

**Static graphs.** Static topologies maintain the same graph connections throughout all iterations. The directedness, degree, and consensus rate of various common topologies are summarized in Table 1. The ring, grid, and torus graphs [22] are the simplest sparse topologies with $\Theta(1)$ maximum degree. However, their consensus rates quickly approach 1 as network size $n$ increases, which leads to inefficient local averaging. The hypercube graph [31] maintains a nice balance between degree and consensus rate since $\ln(n)$ varies slowly with $n$. However, this graph cannot be formed when size $n$ is not the power of 2. The static exponential graph extends hypercubes to graphs with any size $n$, but its directed communications cannot enable symmetric gossip matrices required in well-known decentralized algorithms such as EXTRA [28], Exact-Diffusion [40], NIDS [16], $D^2$ [30]. Two widely-used random topologies, i.e., the Erdos-Renyi graph [20, 3] and the geometric random graph [4, 5], are also listed in Table 1. It is observed that the Erdos-Renyi graph achieves a network-size-independent consensus rate with a $\Theta(\ln(n))$ *averaged* degree in expectation. However, it is worth noting that the communication overhead in network topology is determined by the *maximum* degree. Since some nodes may have much more neighbors than others in a random realization, the maximum degree in the Erdos-Renyi graph can be $O(n)$ with a non-zero probability. Moreover, the random graphs listed in Table 1 are undirected. The may not be used in scenarios where directed graphs are preferred.

**Dynamic graphs.** Dynamic graphs allow time-varying topologies between iterations. When an exponential graph allows each node to cycle through all its neighbors and communicate only to a single node per iteration, we achieve the time-varying one-peer exponential graph [2, 37]. When the network size $n$ is the power of 2, a sequence of one-peer exponential graphs can together achieve periodic global averaging. However, its consensus rate is unknown for other values of $n$. A closely related work Matcha [34] proposed a disjoint matching decomposition sampling strategy when training learning models. While it decomposes a static dense graph into a series of sparse graphs with small degrees, the consensus rates of these dynamic graphs are not established.

Finally, it is worth noting that the consensus rates of all graphs (except for the Erdos-Renyi graph) discussed above are either unknown or dependent on size $n$. Their efficiency in mixing information gets less effective as $n$ goes large.

## 1.2 Main results

**Motivation.** Since existing network topologies suffer from several limitations, we ask the following questions. *Can we develop topologies that have (almost) constant degrees and network-size-independent consensus rates that admit both symmetric and asymmetric matrices of any size? Can these topologies allow one-peer dynamic variants?* This paper provides affirmative answers.

**Main results and contributions.** This paper develops several novel graphs built upon a set of basis graphs in which the label difference between any pair of connected nodes are *equivalent*. With a general name EquiTopo, these new graphs can achieve network-size-independent consensus rates while maintaining (almost) constant graph degrees. Our contributions are:

- We construct a underlined{directed} graph named D-EquiStatic that has a network-size-independent consensus rate $\rho$ with a degree $\Theta(\ln(n))$. Furthermore, we develop an underline{one-peer} time-varying variant named OD-EquiDyn to achieve a network-size-independent consensus rate with degree 1.

- We construct a underlined{undirected} graph U-EquiStatic, which has a network-size-independent consensus rate $\rho$ with degree $\Theta(\ln(n))$. It admits symmetric gossip matrices that are required by various important algorithms. We also develop an underline{one-peer} time-varying and undirected variant named OU-EquiDyn to achieve a network-size-independent consensus rate with degree 1.

- We apply the EquiTopo graphs to two well-known decentralized algorithms, i.e., decentralized stochastic gradient descent (SGD) [6, 17, 12] and stochastic gradient tracking (SGT) [23, 36, 7, 35], to achieve the state-of-the-art convergence rate while maintaining $\Theta(\ln(n))$ (with D/U-EquiStatic) or 1 (with OD/OU-EquiDyn) degree in per-iteration communication.

The comparison between EquiTopo and other common topologies in Table 1 shows that the EquiTopo family (especially the one-peer variants) has achieved the best balance between maximum graph degree and consensus rate. The comparison between EquiTopo and other common topologies when applying to DSGD and DSGT are listed in Tables 3 and 4 in Appendix C.2 and Table 2.

**Note.** This paper considers scenarios in which any two nodes can be connected when necessary. The high-performance data center cluster is one such scenario in which all GPUs are connected with high-bandwidth channels, and the network topology can be fully controlled. EquiTopo may not be applied to wireless network settings where two remote nodes cannot be connected directly.

## 1.3 Other related works

In decentralized optimization, decentralized gradient descent [24, 6, 26, 39] and dual averaging [8] are well-known approaches. While simple and widely used, their solutions are sensitive to heterogeneous data distributions. Advanced algorithms that can overcome this drawback include explicit bias-correction [28, 40, 16], gradient tracking [23, 7, 25, 36], and dual acceleration [27, 33]. Decentralize SGD is extensively studied in [6, 17, 2] to solve stochastic problems. It has been extended to directed [2, 37] or time-varying topologies [12, 21, 34, 37], asynchronous settings [18], and data-heterogeneous scenarios [30, 35, 1, 11, 19] to achieve better performances.

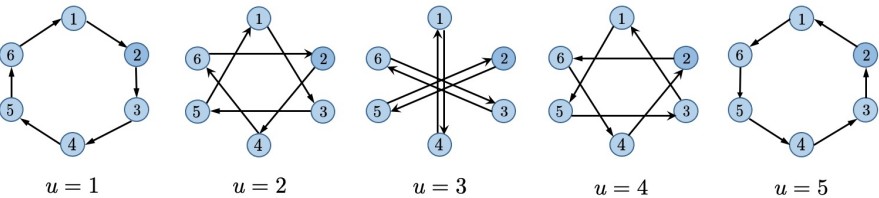

Figure 1: The set of the basis graphs $\{\mathcal{G}(\boldsymbol{A}^{(u)})\}_{u=1}^{5}$ for $n = 6$.

## 2 Notations and Preliminaries

**Notations.** We let $\mathbb{1}_n \in \mathbb{R}^n$ be the all-ones vector and $\boldsymbol{I} \in \mathbb{R}^{n \times n}$ be the identity matrix. Furthermore, we define $\boldsymbol{J} = \frac{1}{n}\mathbb{1}_n\mathbb{1}_n^T$ and $\boldsymbol{\Pi} = \boldsymbol{I} - \boldsymbol{J}$. A matrix $\boldsymbol{A} = [a_{ij}] \in \mathbb{R}^{n \times n}$ is nonnegative if $a_{ij} \geq 0$ for all $1 \leq i, j \leq n$. A nonnegative matrix $\boldsymbol{A}$ is doubly stochastic if $\boldsymbol{A}\mathbb{1}_n = \boldsymbol{A}^T\mathbb{1}_n = \mathbb{1}_n$. Given a matrix $\boldsymbol{A} \in \mathbb{R}^{m \times n}$, $\|\boldsymbol{A}\|_2$ is its spectral norm. For $\boldsymbol{x} \in \mathbb{R}^n$, $\|\boldsymbol{x}\|$ is its Euclidean norm. We let $[n] = \{1, \cdots, n\}$. Throughout the paper, we define a $\mathrm{mod}$ operation that returns a value in $[n]$ as

$$i \bmod n = \begin{cases} \ell & \text{if } i = kn + \ell \text{ for some } k \in \mathbb{Z} \text{ and } \ell \in [n-1], \\ n & \text{if } i = kn \text{ for some } k \in \mathbb{Z}. \end{cases} \tag{1}$$

**Network.** Given a graph $\mathcal{G}(\mathcal{V}, \mathcal{E})$ with a set of $n$ nodes $\mathcal{V}$ and a set of directed edges $\mathcal{E}$. An edge $(j, i) \in \mathcal{E}$ means node $j$ can directly send information to node $i$. For undirected graphs, $(j, i) \in \mathcal{E}$ if and only if $(i, j) \in \mathcal{E}$. Node $i$'s degree is the number of its in-neighbors $|\{j|(j, i) \in \mathcal{E}\}|$. A one-peer graph means that the degree for each node is *at most* 1.

**Weight matrices.** To facilitate the local averaging step in decentralized algorithms, each graph is associated with a nonnegative weight matrix $\boldsymbol{W} = [w_{ij}] \in \mathbb{R}^{n \times n}$, whose element $w_{ij}$ is non-zero only if $(j, i) \in \mathcal{E}$ or $i = j$. One benefit of an undirected graph is that it can be associated with a symmetric matrix. Given a nonnegative weight matrix $\boldsymbol{W} \in \mathbb{R}^{n \times n}$, we let $\mathcal{G}(\boldsymbol{W})$ be its associated graph such that $(j, i) \in \mathcal{E}$ if $w_{ij} > 0$ and $i \neq j$.

**Consensus rate.** For weight matrices $\{\boldsymbol{W}^{(t)}\}_{t \geq 0} \subseteq \mathbb{R}^{n \times n}$, the consensus rate $\beta$ is the minimum nonnegative number such that for any $t \geq 0$ and vector $\boldsymbol{x} \in \mathbb{R}^n$ with the average $\bar{x} = \frac{1}{n}\sum_{i=1}^{n} x_i$,

$$\mathbb{E}\left[\left\|\boldsymbol{W}^{(t)}\boldsymbol{x} - \bar{x}\cdot\mathbb{1}_n\right\|^2\right] \leq \beta^2\left\|\boldsymbol{x} - \bar{x}\cdot\mathbb{1}_n\right\|^2,$$

or equivalently, $\mathbb{E}\left[\|\boldsymbol{\Pi}\boldsymbol{W}^{(t)}\boldsymbol{x}\|^2\right] \leq \beta^2\|\boldsymbol{\Pi}\boldsymbol{x}\|^2$. For $\boldsymbol{W}^{(t)} \equiv \boldsymbol{W}$, $\beta$ essentially equals $\|\boldsymbol{\Pi}\boldsymbol{W}\|_2$.

## 3 Directed EquiTopo Graphs

### 3.1 Basis weight matrices and basis graphs

Given a graph of size $n$, we introduce a set of doubly stochastic *basis matrices* $\{\boldsymbol{A}^{(u,n)}\}_{u=1}^{n-1}$, where $\boldsymbol{A}^{(u,n)} = [a_{ij}^{(u,n)}] \in \mathbb{R}^{n \times n}$ with

$$a_{ij}^{(u,n)} = \begin{cases} \frac{n-1}{n}, & \text{if } i = (j + u) \bmod n, \\ \frac{1}{n}, & \text{if } i = j, \\ 0, & \text{otherwise.} \end{cases} \tag{2}$$

Their associated graphs $\{\mathcal{G}(\boldsymbol{A}^{(u,n)})\}_{u=1}^{n-1}$ are called *basis graphs*. A basis graph $\mathcal{G}(\boldsymbol{A}^{(u,n)})$ has degree one and the same *label difference* $(i - j) \bmod n$ for all edges $(j, i)$. The set of five basis graphs $\{\mathcal{G}(\boldsymbol{A}^{(u,6)})\}_{u=1}^{5}$ for $n = 6$ is shown in Fig. 1. When $n$ is clear from the context, we omit it and write $\boldsymbol{A}^{(u)}$ instead.

### 3.2 Directed static EquiTopo graphs (D-EquiStatic)

Our directed graphs are built on the above basis graphs, and a weight matrix has the form

$$\boldsymbol{W} = \frac{1}{M}\sum_{i=1}^{M}\boldsymbol{A}^{(u_i)}, \tag{3}$$

where $u_i \in [n-1]$ and $M > 0$ is the number of basis graphs we will sample. Throughout this paper, the multiset $\{u_i\}_{i=1}^M$ are called *basis index*. It is possible that $u_i = u_j$ for some $i \neq j$. Since each $\boldsymbol{A}^{(u)}$ has the form (2), the matrix $\boldsymbol{W}$ is doubly stochastic, and all nodes of the directed graph $\mathcal{G}(\boldsymbol{W})$ have the same degree that is no more than $M$.

Since $\mathcal{G}(\boldsymbol{W})$ is a directed static graph and built with $M$ basis graphs, we name it D-EquiStatic. The following theorem shows that we can construct a weight matrix $\boldsymbol{W}$ such that its consensus rate is independent of the network size $n$ by setting $M$ properly. The proofs of all theorems are in the Appendix.

**Theorem 1** *Let $\boldsymbol{A}^{(u)}$ be defined by (2) for any $u \in [n-1]$. For any constant $\rho \in (0,1)$, we can choose a sequence of $u_1, \cdots, u_M$ from $[n-1]$ with $M = \Theta\left(\ln(n)/\rho^2\right)$ and construct the D-EquiStatic weight matrix $\boldsymbol{W}$ as in (3) such that the consensus rate of $\boldsymbol{W}$ is $\rho$, i.e.,*

$$\|\boldsymbol{\Pi}\,\boldsymbol{W}\,\boldsymbol{x}\| \leq \rho\,\|\boldsymbol{\Pi}\boldsymbol{x}\|, \ \forall \boldsymbol{x} \in \mathbb{R}^n \tag{4}$$

The graph $\mathcal{G}(\boldsymbol{W})$ has degree at most $M$. In the following, we will just say that the degree is $\Theta(\ln(n))$. As $\rho$ is tunable, we choose $\rho$ as a constant, e.g., $\rho = 0.5$. A method of constructing D-EquiStatic weight matrix $\boldsymbol{W}$ can be found in Appendix A.1.

**Remark 1** *Compared to all common topologies listed in Table 1, D-EquiStatic achieves a better balance between degree and consensus rate. Moreover, D-EquiStatic works for any size $n \geq 2$. Different from the Erdos-Renyi random graph and the geometric random graph, whose degree cannot be predefined before the implementation, we can easily specify the degree $M$ for D-EquiStatic.*

### 3.3 One-peer directed EquiTopo graphs (OD-EquiDyn)

While D-EquiStatic achieves a size-independent consensus rate with $\Theta(\ln(n))$ degree, we develop a one-peer dynamic variant to further reduce its degree. Given a weight matrix $\boldsymbol{W}$ of form (3) and its associated basis matrix $\{\boldsymbol{A}^{(u_i)}\}_{i=1}^M$, the one-peer directed variant, or OD-EquiDyn for short, samples a random $\boldsymbol{A}^{(u)}$ per iteration and utilizes it as the one-peer weight matrix, see Alg. 1. Since each node in $\mathcal{G}(\boldsymbol{A}^{(u)})$ has exactly one neighbor, the graph $\mathcal{G}(\boldsymbol{W}^{(t)})$ has degree one for every iteration $t$. Note that $\boldsymbol{W}^{(t)}$ is a random time-varying weight matrix. Its consensus rate (in expectation) can be characterized as below.

---
**Algorithm 1:** OD-EquiDyn weight matrix generation at iteration $t$

---
**Input:** constant $\eta \in (0,1)$; basis index $\{u_1, u_2, \ldots, u_M\}$ from a weight matrix $\boldsymbol{W}$ of form (3);
Pick $v_t$ from uniform distribution over the basis index $\{u_1, u_2, \ldots, u_M\}$;
Produce basis matrix $\boldsymbol{A}^{(v_t)}$ according to (2);
**Output:** $\boldsymbol{W}^{(t)} = (1-\eta)\boldsymbol{I} + \eta\boldsymbol{A}^{(v_t)}$

---

**Theorem 2** *Let the one-peer directed weight matrix $\boldsymbol{W}^{(t)}$ be generated by Alg. 1. It holds that*

$$\mathbb{E}\left[\left\|\boldsymbol{\Pi}\,\boldsymbol{W}^{(t)}\boldsymbol{x}\right\|^2\right] \leq \left(1 - 2\eta(1-\eta)(1-\rho)\right)\|\boldsymbol{\Pi}\boldsymbol{x}\|^2, \ \ \forall \boldsymbol{x} \in \mathbb{R}^n$$

*where $\rho$ is the consensus rate of the weight matrix $\boldsymbol{W}$ (which can be tuned freely as in Theorem 1).*

**Remark 2** *The OD-EquiDyn graph has a degree of $1$ no matter how dense the input matrix $\boldsymbol{W}$ is. When $\eta = 1/2$, which is used in our implementations, it holds that $\mathbb{E}\|\boldsymbol{\Pi}\,\boldsymbol{W}^{(t)}\boldsymbol{x}\|^2 \leq (1 + \rho)/2\,\|\boldsymbol{\Pi}\boldsymbol{x}\|^2$ for any $\boldsymbol{x} \in \mathbb{R}^n$. Thus, the OD-EquiDyn graph maintains the same $\Theta(1)$ degree as ring, grid, and the one-peer exponential graph but with a faster size-independent consensus rate.*

**Remark 3** *For basis index $\{1, \cdots, n-1\}$ ($\boldsymbol{W} = \boldsymbol{J}$), Alg. 1 returns a sequence of OD-EquiDyn graphs with $\mathbb{E}\|\boldsymbol{\Pi}\,\boldsymbol{W}^{(t)}\boldsymbol{x}\|^2 \leq \frac{1}{2}\|\boldsymbol{\Pi}\boldsymbol{x}\|^2$ when $\eta = \frac{1}{2}$ because $\|\boldsymbol{\Pi}\boldsymbol{J}\boldsymbol{x}\| = 0$ implies $\rho = 0$.*

**Remark 4** *Although the Erdos-Renyi random graph also enjoys $\Theta(1)$ consensus rate and $\Theta(\ln(n))$ average degree (see Table 1), its maximum degree could be as large as $\Theta(n)$ which implies that*

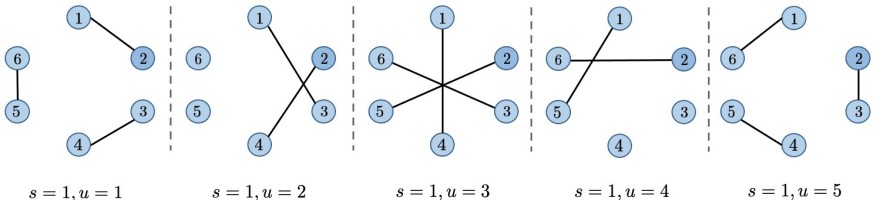

$$s = 1, u = 1 \qquad s = 1, u = 2 \qquad s = 1, u = 3 \qquad s = 1, u = 4 \qquad s = 1, u = 5$$

Figure 2: A few realizations of the OU-EquiDyn graphs for $n = 6$, $s = 1$, and $u \in \{1, 2, 3, 4, 5\}$.

*Erdos-Renyi random graphs could be highly unbalanced. Moreover, Erdos-Renyi random graphs are undirected graphs, while EquiStatic graphs can be both directed (Section 3) and undirected (Section 4). In addition, the structure of EquiStatic allows simple construction of one-peer random graphs which preserve $\Theta(1)$ consensus, while it is still an open problem on whether Erdos-Renyi random graphs admit one-peer variants with $\Theta(1)$ consensus rate.*

## 4 Undirected EquiTopo Graphs

The implementation of many important algorithms such as EXTRA [28], Exact-Diffusion [40], NIDS [16], decentralized ADMM [29], and the dual-based optimal algorithms [27, 33, 13] rely on symmetric weight matrices. Moreover, devices in full-duplex communication systems can communicate with one another in both directions, and undirected networks are natural to be utilized. These motivate us to study undirected graphs.

### 4.1 Undirected static EquiTopo graphs (U-EquiStatic)

Given a D-EquiStatic weight matrix $\boldsymbol{W}$ and its associated basis matrices $\{\boldsymbol{A}^{(u_i)}\}_{i=1}^M$, we directly construct an undirected weight matrix name U-EquiStatic by

$$\widetilde{\boldsymbol{W}} = \tfrac{1}{2}(\boldsymbol{W} + \boldsymbol{W}^T) = \tfrac{1}{2M} \sum_{i=1}^M (\boldsymbol{A}^{(u_i)} + [\boldsymbol{A}^{(u_i)}]^T), \tag{5}$$

whose basis index are $\{u_i, -u_i\}_{i=1}^M$ because $\boldsymbol{A}^{(-u)} = [\boldsymbol{A}^{(u)}]^T$.

Since $\widetilde{\boldsymbol{W}}$ is built upon $\boldsymbol{W}$ and $\boldsymbol{\Pi}\boldsymbol{W} = \boldsymbol{W}\boldsymbol{\Pi}$, the following theorem follows directly from

$$\left\| \boldsymbol{\Pi}\widetilde{\boldsymbol{W}} \right\|_2 = \frac{1}{2} \left\| \boldsymbol{\Pi}\boldsymbol{W} + (\boldsymbol{\Pi}\boldsymbol{W})^T \right\|_2 \leq \frac{1}{2} \left( \|\boldsymbol{\Pi}\boldsymbol{W}\|_2 + \left\| (\boldsymbol{\Pi}\boldsymbol{W})^T \right\|_2 \right) = \|\boldsymbol{\Pi}\boldsymbol{W}\|_2.$$

**Theorem 3** *Let $\boldsymbol{W}$ be a D-EquiStatic matrix with consensus rate $\rho$ and $\widetilde{\boldsymbol{W}}$ be the U-EquiStatic matrix defined by* (5). *It holds that*

$$\left\| \boldsymbol{\Pi}\widetilde{\boldsymbol{W}}\boldsymbol{x} \right\| \leq \rho \|\boldsymbol{\Pi}\boldsymbol{x}\|, \ \ \forall \boldsymbol{x} \in \mathbb{R}^n. \tag{6}$$

### 4.2 One-peer undirected EquiTopo graphs (OU-EquiDyn)

Constructing a one-peer undirected graph OU-EquiDyn is not as direct as U-EquiStatic because $\frac{1}{2}(\boldsymbol{A}^{(u)} + [\boldsymbol{A}^{(u)}]^T)$ admits a graph with degree 2, see Appendix B.1 for an illustration.

Alg. 2 shows a method to construct a series of OU-EquiDyn matrices with degree 1. Starting from node $s$, we connect a node with the $u^{\text{th}}$ node after it, as long as both of them have not been connected to any other nodes. Fig. 2 illustrates the process when $s = 1$ and $u = 1, \cdots, 5$ for a network of size 6. Some nodes have no neighbors at realizations $u = 2$ and $u = 4$. This phenomenon is caused by the restriction that each node has no more than one neighbor. For instance, when $u = 2$, node 5 wants to connect with node 1 but node 1 has already been connected to node 3. Thus, there exist node pairs that are never connected when $s = 1$. To resolve this issue, we let the starting index $s$ be sampled randomly from $[n]$. Fig. 8 in Appendix B.1 illustrates the scenarios when $s = 3$. It is observed that the pairs $\{3, 5\}$ and $\{4, 6\}$ are now connected to each other. The node version of OU-EquiDyn is illustrated in Alg. 4 of Appendix. B.2.

**Algorithm 2:** OU-EquiDyn weight matrix generation at iteration $t$

---

**Input:** $\eta \in (0,1)$; basis index $\{u_i, -u_i\}_{i=1}^M$ from a symmetric weight matrix $\widetilde{W} \in \mathbb{R}^{n \times n}$ of form (5);

Pick $v_t \in \{u_i, -u_i\}_{i=1}^M$ and $s_t \in [n]$ uniformly at random;

Initialize $A = [a_{ij}] = I$ and $b_i = 0, \forall i \in [n]$;

**for** $j = (s_t : s_t + n - 1 \mod n)$ **do**

 $i = (j + v_t) \mod n$;

 **if** $b_i = 0$ *and* $b_j = 0$ **then**

  $a_{ij} = a_{ji} = (n-1)/n$;

  $a_{ii} = a_{jj} = 1/n$;

  $b_i = 1, b_j = 1$;

 **end**

**end**

**Output:** $\widetilde{W}^{(t)} = (1-\eta)I + \eta A$

---

**Theorem 4** *Let $\widetilde{W}$ be a U-EquiStatic matrix with consensus rate $\rho$, and $\widetilde{W}^{(t)}$ be an OU-EquiDyn matrix generated by Alg. 2, it holds that*

$$\mathbb{E}\left[\left\|\Pi\,\widetilde{W}^{(t)}x\right\|^2\right] \leq \left(1 - \frac{4}{3}\eta(1-\eta)(1-\rho)\right)\|\Pi x\|^2, \quad \forall x \in \mathbb{R}^n.$$

**Remark 5** *Theorem 4 implies that the OU-EquiDyn graph can achieve a size-independent consensus rate with a degree at most 1. When $\eta = 1/2$, it holds that $\mathbb{E}\|\Pi W^{(t)}x\|^2 \leq [(2+\rho)/3]\|\Pi x\|^2$.*

When $\widetilde{W} = J$ and the basis index $\{1, \cdots, n-1\}$ are input to Alg. 2, we obtain an OU-EquiDyn sequence $\widetilde{W}^{(t)}$ such that $\mathbb{E}\|\Pi\,\widetilde{W}^{(t)}x\|^2 \leq (2/3)\|\Pi x\|^2$.

**Remark 6** *An alternative OU-EquiDyn matrix construction that relies on the Euclidean algorithm is in Appendix B.4.*

## 5 Applying EquiTopo Matrices to Decentralized Learning

We consider the following distributed problem over a network of $n$ computing nodes:

$$\min_{x \in \mathbb{R}^d} \quad f(x) = \frac{1}{n}\sum_{i=1}^n f_i(x) \tag{7}$$

where $f_i(x) := \mathbb{E}_{\xi_i \sim \mathcal{D}_i}[F(x; \xi_i)]$. The function $f_i(x)$ is kept at node $i$, and $\xi_i$ denotes the local data that follows the local distribution $\mathcal{D}_i$. Data heterogeneity exists if local distributions $\{\mathcal{D}_i\}_{i=1}^n$ are not identical. Throughout this section, we let $x_i^{(t)}$ be node $i$'s local model at iteration $t$, and $\bar{x}^{(t)} = \frac{1}{n}\sum_{i=1}^n x_i^{(t)}$.

**Assumptions.** We make the following standard assumptions to facilitate analysis.

**A.1** *Each local cost function $f_i(x)$ is differentiable, and there exists a constant $L > 0$ such that $\|\nabla f_i(x) - \nabla f_i(y)\| \leq L\|x - y\|$ for all $x, y \in \mathbb{R}^d$.*

**A.2** *Let $g_i^{(t)} = \nabla F(x_i^{(t)}; \xi_i^{(t)})$. There exists $\sigma^2 > 0$ such that for any $t$ and $i$*

$$\mathbb{E}_{\xi_i^{(t)} \sim \mathcal{D}_i} g_i^{(t)} = \nabla f_i(x_i^{(t)}), \quad \text{and} \quad \mathbb{E}_{\xi_i^{(t)} \sim \mathcal{D}_i}\left[\left\|g_i^{(t)} - \nabla f_i(x_i^{(t)})\right\|^2\right] \leq \sigma^2.$$

**A.3** *(For DSGD only) There exists $b^2$ such that $\frac{1}{n}\sum_{i=1}^n \|\nabla f_i(x) - \nabla f(x)\| \leq b^2$ for all $x \in \mathbb{R}^d$.*

### 5.1 Decentralized stochastic gradient descent

The decentralized stochastic gradient descent (DSGD) [6, 17, 12] is given by

$$x_i^{(t+1)} = \sum_{j=1}^n w_{ij}^{(t)}\left(x_j^{(t)} - \gamma g_j^{(t)}\right), \tag{8}$$

where the weight matrix $\boldsymbol{W}^{(t)} = [w_{ij}^{(t)}]$ can be time-varying and random. Applying the EquiTopo matrices discussed in § 3-4, we achieve the following convergence results, whose proof follows Theorem 2 in [12] directly, and is omitted here. More results for DSGD are given in Appendix C.1.

**Theorem 5** *Consider the DSGD algorithm* (8). *Under Assumptions A.1-A.3, it holds that*

$$\frac{1}{T+1}\sum_{t=0}^{T}\mathbb{E}\Big[\left\|\nabla f(\bar{\boldsymbol{x}}^{(t)})\right\|^2\Big] = \mathcal{O}\Big(\frac{\sigma}{\sqrt{nT}} + \frac{\beta^{\frac{2}{3}}\sigma^{\frac{2}{3}}}{T^{\frac{2}{3}}(1-\beta)^{\frac{1}{3}}} + \frac{\beta^{\frac{2}{3}}b^{\frac{2}{3}}}{T^{\frac{2}{3}}(1-\beta)^{\frac{2}{3}}} + \frac{\beta}{T(1-\beta)}\Big),$$

- *where $\beta = \rho$ with D-EquiStatic $\boldsymbol{W}$ or U-EquiStatic $\widetilde{\boldsymbol{W}}$;*

- *where $\beta = \sqrt{(1+\rho)/2}$ for OD-EquiDyn $\boldsymbol{W}^{(t)}$ (Alg. 1 with $\eta = 1/2$), and $\beta = \sqrt{(2+\rho)/3}$ for OU-EquiDyn $\widetilde{\boldsymbol{W}}^{(t)}$ (Alg. 2 with $\eta = 1/2$).*

For a sufficiently large $T$, the term $\mathcal{O}(1/\sqrt{nT})$ dominates the rate, and we say the algorithm reaches the linear speedup stage. The transient iterations are referred to as those iterations before an algorithm reaches the linear-speedup stage. We compare the per-iteration communication, convergence rate, and transient iterations of DSGD over various topologies in Tables 3 and 4 of the Appendix. It is observed that OD/OU-EquiDyn endows DSGD with the lightest communication, fastest convergence rate, and smallest transient iteration complexity.

## 5.2 Decentralized stochastic gradient tracking algorithm

The decentralized stochastic gradient tracking algorithm (DSGT) [23, 7, 25, 36, 35] is given by

$$\boldsymbol{x}_i^{(t+1)} = \sum_{j=1}^{n} w_{ij}^{(t)}\big(\boldsymbol{x}_j^{(t)} - \gamma\boldsymbol{y}_j^{(t)}\big);$$
$$\boldsymbol{y}_i^{(t+1)} = \sum_{j=1}^{n} w_{ij}^{(t)}\boldsymbol{y}_j^{(t)} + \boldsymbol{g}_i^{(t+1)} - \boldsymbol{g}_i^{(t)}, \quad \boldsymbol{y}_i^{(0)} = \boldsymbol{g}_i^{(0)}. \tag{9}$$

The following result of DSGT does not appear in the literature since it admits an improved convergence rate for stochastic decentralized optimization over asymmetric or time-varying weight matrices. Existing works on DSGT assume weight matrix to be either symmetric [11, 1] or static [35].

**Theorem 6** *Consider the DSGT algorithm in* (9). *If* $\{\boldsymbol{W}^{(t)}\}_{t\geq 0}$ *have consensus rate $\beta$, then under Assumptions A.1-A.2, it holds for $T \geq \frac{1}{1-\beta}$ that*

$$\frac{1}{T+1}\sum_{t=0}^{T}\mathbb{E}\Big[\|\nabla f(\bar{\boldsymbol{x}}^{(t)})\|^2\Big] = \mathcal{O}\Big(\frac{\sigma}{\sqrt{nT}} + \frac{\sigma^{\frac{2}{3}}}{(1-\beta)T^{\frac{2}{3}}} + \frac{1}{(1-\beta)^2 T}\Big).$$

When utilizing the EquiTopo matrices, the corresponding $\beta$ is specified in Theorem 5. Note that DSGT achieves linear speedup for large $T$. The per-iteration communication and convergence rate comparison of DSGT over different topologies is in Table 2. OD/OU-EquiDyn endows DSGT with the lightest communication, fastest convergence rate, and smallest transient iteration complexity.

## 6 Numerical Experiments

This section presents experimental results to validate EquiTopo's network-size-independent consensus rate and its comparison with other commonly-used topologies in DSGD on both strongly-convex problems and non-convex deep learning tasks. More experiments for EquiTopo in DSGT and all implementation details are referred to Appendix D.

**Network-size independent consensus rate.** This simulation examines the consensus rates of all four EquiTopo graphs. We recursively run the gossip averaging $\boldsymbol{x}^{(t+1)} = \boldsymbol{W}^{(t)}\boldsymbol{x}^{(t)}$ with $\boldsymbol{x}^{(0)} \in \mathbb{R}^n$ initialized arbitrarily and $\boldsymbol{W}^{(t)} \in \mathbb{R}^{n \times n}$ generated as D-EquiStatic (Eq. (3)), OD-EquiDyn (Alg. 1), U-EquiStatic (Eq. (5)), and OU-EquiDyn (Alg. 2), respectively. Fig. 3 depicts how the quantity $\|\boldsymbol{x}^{(t)} - \boldsymbol{J}\boldsymbol{x}^{(0)}\|$ evolves when $n$ ranges from 1000 to 10,000 with D-EquiStatic topology. See Appendix D for all other EquiTopo graphs, which also achieve network-size independent consensus rates when $n$ varies. These results are consistent with Theorems 1 - 4.

Table 2: Per-iteration communication and computation complexity of the DSGT under different topologies.

| Topology | Per-iter Comm. | Convergence Rate | Trans. Iters. |
|---|---|---|---|
| Ring | $\Theta(1)$ | $\mathcal{O}\left(\frac{\sigma}{\sqrt{nT}} + \frac{n^2\sigma^{\frac{2}{3}}}{T^{\frac{2}{3}}} + \frac{n^4}{T}\right)$ | $\mathcal{O}(n^{15})$ |
| Torus | $\Theta(1)$ | $\mathcal{O}\left(\frac{\sigma}{\sqrt{nT}} + \frac{n\sigma^{\frac{2}{3}}}{T^{\frac{2}{3}}} + \frac{n^2}{T}\right)$ | $\mathcal{O}(n^9)$ |
| Static Exp. | $\Theta(\ln(n))$ | $\mathcal{O}\left(\frac{\sigma}{\sqrt{nT}} + \frac{\ln(n)\sigma^{\frac{2}{3}}}{T^{\frac{2}{3}}} + \frac{\ln^2(n)}{T}\right)$ | $\mathcal{O}(n^3\ln^6(n))$ |
| O.-P. Exp. | $1$ | $\mathcal{O}\left(\frac{\sigma}{\sqrt{nT}} + \frac{\ln(n)\sigma^{\frac{2}{3}}}{T^{\frac{2}{3}}} + \frac{\ln^2(n)}{T}\right)$ | $\mathcal{O}(n^3\ln^6(n))$ |
| D(U)-EquiStatic | $\Theta(\ln(n))$ | $\mathcal{O}\left(\frac{\sigma}{\sqrt{nT}} + \left(\frac{\sigma}{T}\right)^{\frac{2}{3}} + \frac{1}{T}\right)$ | $\mathcal{O}(n^3)$ |
| OD (OU)-EquiDyn | $1$ | $\mathcal{O}\left(\frac{\sigma}{\sqrt{nT}} + \left(\frac{\sigma}{T}\right)^{\frac{2}{3}} + \frac{1}{T}\right)$ | $\mathcal{O}(n^3)$ |

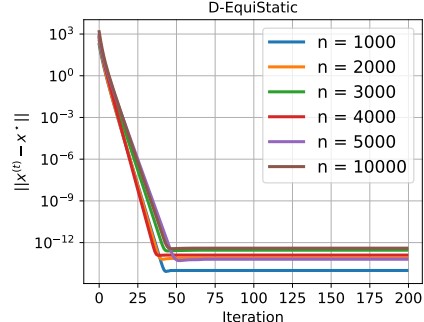

Figure 3: The D-EquiStatic topology can achieve network-size independent consensus rate.

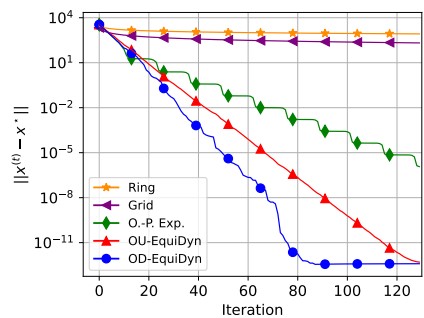

Figure 4: OD/OU-EquiDyn is faster than other topologies (i.e., ring, grid, and one-peer exponential graph) with $\Theta(1)$ degree in consensus rate.

**Comparison with other topologies.** We now compare EquiTopo's consensus rate with other commonly-used topologies. Fig. 4 illustrates the performance of several graphs with $\Theta(1)$ degree when running gossip averaging. We set $n = 4900$ so that the grid graph can be organized as $70 \times 70$. OD/OU-EquiDyn is much faster than other topologies. Note that each node in OD-EquiDyn, OU-EquiDyn, and O.-P. Exp. has exactly one neighbor per iteration. More experiments on graphs with $\Theta(\ln(n))$ degrees and on scenarios with smaller network sizes are in Appendix D.

**DSGD with EquiTopo: least-square.** We next apply D/U-EquiStatic graphs to DSGD when solving the distributed least square problems. In the experiment, we let $n = 300$ and set $M = 9$ so that D/U-EquiStatic has the same degree as the exponential graph. Fig. 5 depicts that D/U-EquiStatic converges much faster than a static exponential graph, especially in the initial stages when the learning rate is large. The U-EquiStatic performs slightly better than D-EquiStatic since its bi-directional communication enables the graph with better connectivity.

**DSGD with EquiTopo: deep learning.** We consider the image classification task with ResNet-20 model [9] over the CIFAR-10 dataset [14]. We utilize BlueFog [38] to support decentralized communication and topology setting in a cluster of 17 Tesla P100 GPUs. Fig. 6 illustrates how D/U-EquiStatic compares with static exp., ring, and centralized SGD in training loss and test accuracy. It is observed that D/U-EquiStatic has strong performance. They achieve competitive training losses to centralized SGD but slightly better test accuracy. Meanwhile, they also outperform static exponential graphs in test accuracy by a visible margin (D-EquiStatic: 92%, U-EquiStatic: 91.7%, Static Exp.: 91.5%). Experiments with EquiDyn topologies and results on MNIST dataset [15] are in Appendix D.

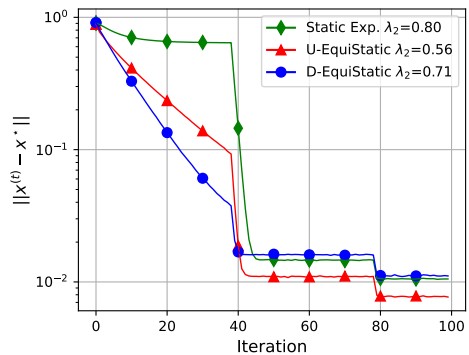

Figure 5: D/U-EquiStatic in DSGD. $\lambda_2$ is the second largest eigenvalue.

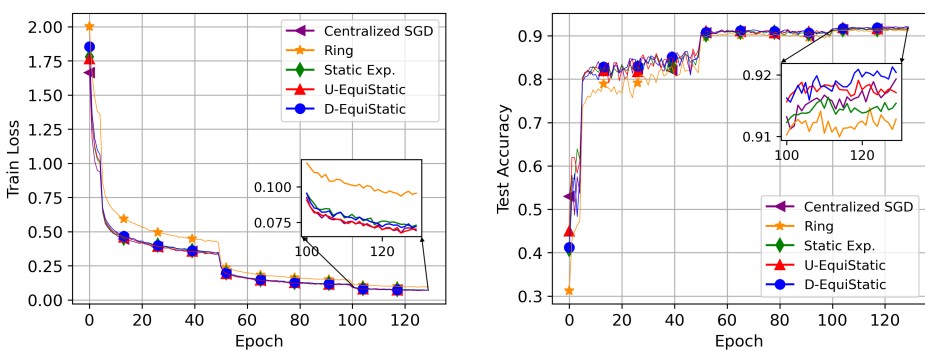

Figure 6: Train loss and test accuracy comparisons among different topologies for ResNet-20 on CIFAR-10.

## 7  Conclusion

This paper proposes EquiTopo graphs that achieve a state-of-the-art balance between the maximum degree and consensus rate. The EquiStatic graphs are with $\Theta(\ln(n))$ degrees and $n$-independent consensus rates, while their one-peer variants, EquiDyn, maintain roughly the same consensus rates with a degree at most 1. EquiTopo enables decentralized learning with light communication and fast convergence. To the best of our knowledge, we are not aware of any **negative social impacts** in our results.

## Acknowledgement

We thank the anonymous reviewers for suggesting updated literature on E.-R. graphs. Ming Yan was partially supported by the NSF award DMS-2012439. Lei Shi was partially supported by Shanghai Science and Technology Program under Project No. 21JC1400600 and No. 20JC1412700 and National Natural Science Foundation of China (NSFC) under Grant No. 12171093. Kexin Jin was supported by Alibaba Research Internship Program.

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
