# A Directed EquiTopo Graphs

## A.1 Construction of a D-EquiStatic graph

A practical method to construct a D-EquiStatic weight matrix $\boldsymbol{W}$ is provided in Alg. 3. We should mention that the "while" loop in the algorithm is adopted to guarantee $\|\boldsymbol{\Pi}\,\boldsymbol{W}\|_2 \leq \rho$.

---

**Algorithm 3:** A practical method for D-EquiStatic weight matrix generation

**Input:** Network size $n$; desired consensus rate $\rho \in (0, 1)$; probability $p$

Set $M = \left\lceil \frac{8}{3\rho^2} \ln(2n/p) \right\rceil$ and initialize $\boldsymbol{W} = \boldsymbol{I}$;

**while** $\|\boldsymbol{\Pi}\,\boldsymbol{W}\|_2 > \rho$ **do**

> Sample $M$ i.i.d random variables $u_1, u_2, \ldots, u_M$ uniformly from $[n-1]$;
> Generate basis weight matrices $\{A^{(u_i)}\}_{i=1}^M$ according to (2);
> Construct $\boldsymbol{W}$ by (3);

**end**

**Output:** The D-EquiStatic weight matrix $\boldsymbol{W}$ and its associated basis indices $\{u_t\}_{t=1}^M$

---

## A.2 Proof of Theorem 1

Before showing properties of $\boldsymbol{W}$ defined by (3), we provide two lemmas as follows.

Referring to Theorem 1.6 of [32], we have the following result for a sequence of random matrices.

**Lemma 1 (Matrix Bernstein)** *Consider a sequence of $K$ independent random $n \times n$ matrices $\{\boldsymbol{M}_i\}_{i=1}^K$. Assume that each random matrix satisfies*

$$\mathbb{E}[\boldsymbol{M}_i] = 0, \quad \text{and} \quad \|\boldsymbol{M}_i\|_2 \leq R \text{ almost surely.}$$

*Define*

$$\sigma^2 := \max\left\{ \left\| \sum_{i=1}^K \mathbb{E}[\boldsymbol{M}_i\boldsymbol{M}_i^T] \right\|_2, \left\| \sum_{i=1}^K \mathbb{E}[\boldsymbol{M}_i^T\boldsymbol{M}_i] \right\|_2 \right\}.$$

*It holds that*

$$\mathbb{P}\left( \left\| \sum_{i=1}^K \boldsymbol{M}_i \right\|_2 \geq \delta \right) \leq 2n \exp\left( -\frac{\delta^2/2}{\sigma^2 + R\delta/3} \right), \quad \forall \delta \geq 0.$$

**Lemma 2** *For any matrix $\boldsymbol{B} = [b_{ij}] \in \mathbb{R}^{n \times n}$, it holds that*

$$\|\boldsymbol{B}\|_2 \leq \max\{\|\boldsymbol{B}\|_1, \|\boldsymbol{B}\|_\infty\}.$$

*Proof.* By definition,

$$\|\boldsymbol{B}\|_2 = \sup_{\|\boldsymbol{x}\| \leq 1, \|\boldsymbol{y}\| \leq 1} \boldsymbol{x}^T \boldsymbol{B} \boldsymbol{y}.$$

Moreover, for all $\|\boldsymbol{x}\| \leq 1$, $\|\boldsymbol{y}\| \leq 1$, we have

$$(\boldsymbol{x}^T \boldsymbol{B} \boldsymbol{y})^2 \leq \left( \sum_{i,j} |b_{ij}| x_i^2 \right)\left( \sum_{i,j} |b_{ij}| y_j^2 \right) \leq \|\boldsymbol{B}\|_1 \|\boldsymbol{B}\|_\infty.$$

Thus, Lemma 2 holds. $\qquad\square$

**Theorem 1** *(Formal restatement of Theorem 1) Let $\boldsymbol{A}^{(u)}$ be defined by (2) for any $u \in [n-1]$ and the D-EquiStatic weight matrix $\boldsymbol{W}$ be constructed by (3) with $\{u_i\}_{i=1}^M$ following an independent and identical uniform distribution from $[n-1]$. For any size-independent consensus rate $\rho \in (0, 1)$ and probability $p \in (0, 1)$, if $M \geq \frac{8}{3\rho^2} \ln \frac{2n}{p}$ it holds with probability at least $1 - p$ that*

$$\|\boldsymbol{\Pi}\,\boldsymbol{W}\,\boldsymbol{x}\| \leq \rho \|\boldsymbol{\Pi}\boldsymbol{x}\|, \ \forall \boldsymbol{x} \in \mathbb{R}^n. \tag{10}$$

*Proof.* Notice that
$$\mathbb{E}[\boldsymbol{A}^{(u_i)}] = \boldsymbol{J}, \ \ \forall i \in \{1, 2, \ldots, M\}.$$
Since each $\boldsymbol{A}^{(u_i)}$ is doubly stochastic, it follows from Lemma 2 that
$$\left\| \boldsymbol{A}^{(u_i)} - \boldsymbol{J} \right\|_2 = \left\| \boldsymbol{\Pi} \boldsymbol{A}^{(u_i)} \right\|_2 \le \|\boldsymbol{\Pi}\|_2 \left\| \boldsymbol{A}^{(u_i)} \right\|_2 \le 1.$$
Consequently,
$$\sum_{i=1}^{M} \mathbb{E}\Big[ \left\| (\boldsymbol{A}^{(u_i)} - \boldsymbol{J})(\boldsymbol{A}^{(u_i)} - \boldsymbol{J})^T \right\|_2 \Big] \le \sum_{i=1}^{M} \mathbb{E}\Big[ \left\| \boldsymbol{\Pi} \boldsymbol{A}^{(u_i)} \right\|_2^2 \Big] \le M.$$

Analogously, $\sum_{i=1}^{M} \mathbb{E}\Big[ \left\| (\boldsymbol{A}^{(u_i)} - \boldsymbol{J})^T (\boldsymbol{A}^{(u_i)} - \boldsymbol{J}) \right\|_2 \Big] \le M$. By Lemma 1,

$$\mathbb{P}\big( \|\boldsymbol{W} - \boldsymbol{J}\|_2 \ge \rho \big) = \mathbb{P}\Big( \Big\| \sum_{i=1}^{M}(\boldsymbol{A}^{(u_i)} - \boldsymbol{J}) \Big\|_2 \ge M\rho \Big)$$
$$\le 2n \exp\Big( -\frac{M^2 \rho^2/2}{M + M\rho/3} \Big) \le 2n \exp\Big( -\frac{M^2 \rho^2/2}{M + M/3} \Big) \le p,$$

i.e.,
$$\mathbb{P}(\|\boldsymbol{\Pi}\boldsymbol{W}\|_2 \le \rho) \ge 1 - p.$$
Note that $\boldsymbol{\Pi}\boldsymbol{W} = \boldsymbol{\Pi}\boldsymbol{W}\boldsymbol{\Pi}$. If $\|\boldsymbol{\Pi}\boldsymbol{W}\|_2 \le \rho$, then
$$\|\boldsymbol{\Pi}\boldsymbol{W}\boldsymbol{x}\|^2 = \|\boldsymbol{\Pi}\boldsymbol{W}\boldsymbol{\Pi}\boldsymbol{x}\|^2 \le \|\boldsymbol{\Pi}\boldsymbol{W}\|_2^2 \|\boldsymbol{\Pi}\boldsymbol{x}\|^2 \le \rho^2 \|\boldsymbol{\Pi}\boldsymbol{x}\|^2.$$
Therefore, the conclusion holds. $\qquad\square$

The relation $M \ge \frac{8}{3\rho^2}\ln(2n/p)$ is required for theoretical analysis, and it is very conservative. In practice, we can set $M$ to be far less than $\frac{8}{3\rho^2}\ln(2n/p)$ and repeat the process described in the formal version of Theorem 1 until we find a desirable $\boldsymbol{W}$ (see Alg. 3). In addition, the verification condition $\|\boldsymbol{\Pi}\boldsymbol{W}\|_2 \le \rho$ in Alg. 3 can also be dropped in implementations so that we only conduct the "while" loop once. We find that these relaxations can still achieve $\boldsymbol{W}$ with an empirically fast consensus rate (see the illustration in the experiments).

**Remark 7** *We have much flexibility in the choice of p, such as $p = 1/2$ or $p = 1/n$. If $p \in (1/\mathrm{poly\,(n)}, 1)$, the corresponding value of M will only differ in constants.*

### A.3 Proof of Theorem 2

Due to $\boldsymbol{\Pi}\boldsymbol{A}^{(v_t)} = \boldsymbol{A}^{(v_t)}\boldsymbol{\Pi} = \boldsymbol{\Pi}\boldsymbol{A}^{(v_t)}\boldsymbol{\Pi}$, it follows from Alg. 1 that
$$\mathbb{E}\Big[ \left\| \boldsymbol{\Pi}\boldsymbol{W}^{(t)}\boldsymbol{x} \right\|^2 \Big] = \mathbb{E}\Big[ \left\| \boldsymbol{\Pi}\Big( (1-\eta)\boldsymbol{I} + \eta\boldsymbol{A}^{(v_t)} \Big)\boldsymbol{x} \right\|^2 \Big]$$
$$\le (1-\eta)^2 \|\boldsymbol{\Pi}\boldsymbol{x}\|^2 + 2\eta(1-\eta)(\boldsymbol{\Pi}\boldsymbol{x})^T \mathbb{E}\big[ \boldsymbol{\Pi}\boldsymbol{A}^{(v_t)} \big] \boldsymbol{\Pi}\boldsymbol{x} + \eta^2 \mathbb{E}\Big[ \left\| \boldsymbol{A}^{(v_t)} \right\|_2^2 \Big] \|\boldsymbol{\Pi}\boldsymbol{x}\|^2$$
Notice that $\mathbb{E}\big[ \boldsymbol{A}^{(v_t)} \big] = \boldsymbol{W}$ and $\|\boldsymbol{\Pi}\boldsymbol{W}\boldsymbol{y}\| \le \rho\|\boldsymbol{\Pi}\boldsymbol{y}\|, \forall \boldsymbol{y} \in \mathbb{R}^n$. Therefore,
$$\mathbb{E}\Big[ \left\| \boldsymbol{\Pi}\boldsymbol{W}^{(t)}\boldsymbol{x} \right\|^2 \Big] \le \big( (1-\eta)^2 + 2\eta\rho(1-\eta) + \eta^2 \big) \|\boldsymbol{\Pi}\boldsymbol{x}\|^2$$
$$= \big( 1 - 2\eta(1-\eta)(1-\rho) \big) \|\boldsymbol{\Pi}\boldsymbol{x}\|^2.$$
The proof is completed. $\qquad\square$

## B One-Peer Undirected EquiTopo Graphs (OU-EquiDyn)

### B.1 Illustration for basis graphs of OU-EquiDyn

The associated graph of $\widehat{\boldsymbol{A}}^{(u)} = \frac{1}{2}(\boldsymbol{A}^{(u)} + [\boldsymbol{A}^{(u)}]^T)$ with $n = 6$ are given in Fig. 7. Clearly, there exist non-one-peer graphs.

Consider $n = 6$. The OU-EquiDyn graphs generated by Alg. 2 (when $s = 3$) are presented in Fig. 8.

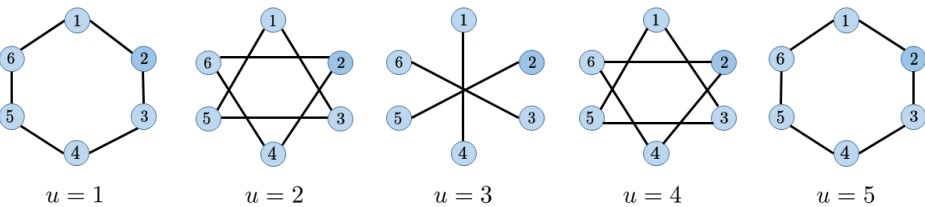

Figure 7: Undirected graphs generated by $\widehat{\boldsymbol{A}}^{(u)} = \frac{1}{2}(\boldsymbol{A}^{(u)} + [\boldsymbol{A}^{(u)}]^T)$.

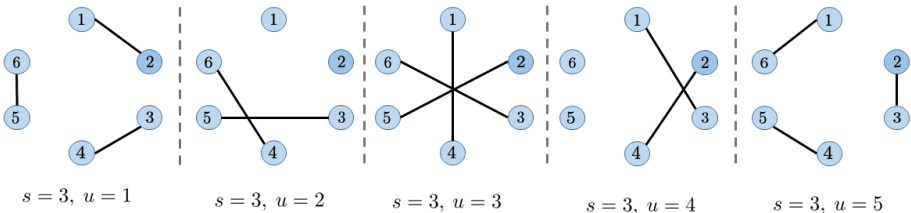

Figure 8: One-peer undirected graphs generated in Alg. 2 with $s = 3$ and $u = 1, \ldots, 5$.

## B.2 Node version of Alg. 2

From the node's perspective, an equivalent version of Alg. 2 is presented in Alg. 4. In the remainder of Appendix B, we denote $\lfloor x \rfloor$ as the largest integer no greater than $x$. We also denote the traditional mod operator as

$$\text{mod}(a, b) = a - b \cdot \left\lfloor \frac{a}{b} \right\rfloor .$$

If $v_t = n/2$, every node $i$ is connected to node $(i + n/2) \mod n$ for Alg. 2, which is the same as Alg. 4.

If $v_t < \frac{n}{2}$, then $q = v_t$. In Alg. 2, nodes $\{s_t, s_t + 1, \cdots, s_t + q - 1\} \mod n$ are connected to $\{s_t + q, s_t + q + 1, \cdots, s_t + 2q - 1\} \mod n$, respectively. If $j \in \{s_t + q, s_t + q + 1, \cdots, s_t + 2q - 1\} \mod n$, then new edges cannot be added because they have been connected. Similar process starts from connecting node $(s_t + 2q) \mod n$ with node $(s_t + 3q) \mod n$, and as a result, for $v_t < \frac{n}{2}$, Alg. 2 can be interpreted as follows: divide $[n]$ into $q$ disjoint subsets:

$$C_\ell = \{i \in [n] : i = (s_t + \ell + d \cdot q) \mod n, \, d \in \mathbb{Z}\}, \, 0 \le \ell < q.$$

Equivalently,

$$C_\ell = \left\{i \in [n] : i = (s_t + \ell + d \cdot q) \mod n, \, 0 \le d \le \left\lfloor \frac{n - 1 - \ell}{q} \right\rfloor \right\} .$$

For node $i$, we define $k(i) = \text{mod}(i - s_t, n)$, $d(i) = \lfloor k(i)/q \rfloor$ and $r(i) = \text{mod}(k(i), q)$. Then, $i \in C_{r(i)}$ and $i = (s_t + r(i) + d(i) \cdot q) \mod n$.

In each $C_\ell$, $s_t + \ell$ is connected with $s_t + \ell + q$, $s_t + \ell + 2q$ is connected with $s_t + \ell + 3q$, $\ldots$. Thus, if $|C_\ell|$ is even, every node in $C_\ell$ has a neighbor. If $|C_\ell|$ is odd (equivalently, $\left\lfloor \frac{n-1-\ell}{q} \right\rfloor$ is odd), the node $\ell + q \cdot \left\lfloor \frac{n-1-\ell}{q} \right\rfloor$ is idle and the others has neighbors.

Define

$$C'_\ell = \begin{cases} C_\ell, & \text{if } \left\lfloor \frac{n-1-\ell}{q} \right\rfloor \text{ is odd} \\ \left\{i \in [n] : i = (s_t + \ell + d \cdot q) \mod n, \, 0 \le d < \left\lfloor \frac{n - 1 - \ell}{q} \right\rfloor \right\}, & \text{otherwise.} \end{cases}$$

Then, node $i$ has a neighbor if and only if it is in the set $C'_{r(i)}$. In addition, for node $i$ in $C'_{r(i)}$, it is connected to $(i + q) \mod n$ if $d(i)$ is even; and connected to $(i - q) \mod n$ if $d(i)$ is odd.

**Algorithm 4:** OU-EquiDyn weight matrix generation at iteration $t$ (from nodes' perspective)

---

**Input:** $\eta \in (0,1)$; basis index $\{u_i, -u_i\}_{i=1}^{M}$ from a weight matrix $\widetilde{\boldsymbol{W}} \in \mathbb{R}^{n \times n}$ of form (5);

Initialize $\boldsymbol{A} = [a_{ij}] = \boldsymbol{I}$;

**for** *node $i = 1$ to $n$ (in parallel)* **do**

    Pick $v_t$ from $\{u_i, -u_i\}_{i=1}^{M}$ and $s_t \in [n]$ uniformly at random using the common random seed;

    **if** $v_t \leq n/2$ **then**

        $q = v_t$;

        $k(i) = \mathrm{mod}(i - s_t, n)$;

    **end**

    **else**

        $q = n - v_t$;

        $k(i) = \mathrm{mod}(i - s_t + q, n)$;

    **end**

    $r(i) = \mathrm{mod}(k(i), q)$;

    $d(i) = \lfloor k(i)/q \rfloor$;

    **if** $\lfloor (n - 1 - r(i))/q \rfloor$ *is odd or* $d(i) < \lfloor (n - 1 - r(i))/q \rfloor$ **then**

        **if** $d(i)$ *is even* **then**

            $j = (i + q) \bmod n$;

        **end**

        **else**

            $j = (i - q) \bmod n$;

        **end**

        $a_{ij} = (n - 1)/n$;

        $a_{ii} = 1/n$;

    **end**

**end**

**Output:** $\widetilde{\boldsymbol{W}}^{(t)} = (1 - \eta)\boldsymbol{I} + \eta\boldsymbol{A}$

---

In Alg. 4, we compute $r(i)$ and $d(i)$ firstly, and then check whether node $i$ is in $C'_{r(i)}$. If $i \in C'_{r(i)}$ and $d(i)$ is even (odd), then it is connected to $(i + q) \bmod n$ $((i - q) \bmod n)$. Otherwise, node $i$ is idle, i.e., $a_{ii} = 1$. This yields the equivalence between Alg.2 and Alg. 4 for the case of $v_t < n/2$.

If $v_t > n/2$, let $q = n - v_t < n/2$. Then the nodes in $\{s_t, s_t + 1, \cdots, s_t + q - 1\} \mod n$ are connected to the nodes $\{s_t + v_t, s_t + 1 + v_t, \cdots, s_t + q - 1 + v_t\} \mod n$ respectively. Note that $(i + v_t) \bmod n$ is equivalent to $(i - q) \bmod n$. Then, equivalently, nodes $\{s_t, s_t + 1, \cdots, s_t + q - 1\} \mod n$ are connected with $\{s_t - q, s_t - q + 1, \cdots, s_t - 1\} \mod n$, respectively. Similar process starts from connecting node $(s_t + 2q) \bmod n$ with node $(s_t + q) \bmod n$. Then, the undirected graph generated with the starting point $s_t$ and the label difference $v_t > n/2$ is equivalent to the undirected graph generated with the starting point $(s_t - q) \bmod n$ and the label difference $q$. So by setting $k(i) = \mathrm{mod}(i - (s_t - q), n)$, the proof follows by similar arguments for the case $v_t < n/2$.

### B.3   Proof of Theorem 4

We first provide the following three lemmas. In the remainder of Appendix B, for any matrix $\boldsymbol{A} = [a_{ij}] \in \mathbb{R}^{n \times n}$, we denote its edge set as

$$\mathcal{E}(\boldsymbol{A}) = \{(i, j) \in [n] \times [n] : a_{ij} > 0, \ i \neq j\}.$$

Denote the matrix $\boldsymbol{A}$ generated at the $t$-th iteration of Alg. 2 by $\widetilde{\boldsymbol{A}}^{(v_t)} = [\widetilde{a}_{ij}^{(v_t)}] \in \mathbb{R}^{n \times n}$. Note that $\widetilde{\boldsymbol{A}}^{(v_t)}$ is also stochastic even when $v_t$ is given since $s_t$ is randomly chosen from $[n]$.

**Lemma 3** *For any $n \geq 2$, it holds for $\widetilde{A}^{(v_t)}$ defined by Alg. 2 that it holds that*

$$\mathbb{E}\big[|\mathcal{E}(\widetilde{A}^{(v_t)})|\big] \geq \frac{2n}{3}.$$

*Proof.* Because $|\mathcal{E}(\widetilde{A}^{(v_t)})|$ is invariant with respect to $s_t$, it suffices to prove $|\mathcal{E}(\widetilde{A}^{(v_t)})| \geq 2n/3$ for $s_t = 1$.

For $v_t \leq n/2$, we define $m = \lfloor n/(2v_t) \rfloor$ and $r = \mathrm{mod}(n, 2v_t)$, then, $m \geq 1$. Notice that node $i$ is connected with $i + v_t$ for any $i \in \{\ell + 2dv_t : 1 \leq \ell \leq v_t,\ 0 \leq d < m\}$.

If $r \leq v_t$, then the last $r$ nodes are idle. As $m \geq 1$, we have $n = 2v_t m + r \geq 2v_t + r \geq 3r$. Thus,

$$|\mathcal{E}(\widetilde{A}^{(v_t)})| \geq n - r \geq n - \frac{1}{3}n = \frac{2}{3}n.$$

If $r > v_t$, then node $i$ in $\{2mv_t + \ell : 1 \leq \ell \leq r - v_t\}$ is connected to $i + v_t$. As a result, only the nodes in $\{2mv_t + \ell : r - v_t + 1 \leq \ell \leq v_t\}$ are idle, i.e., $2v_t - r$ nodes are idle. We have $n = 2mv_t + r > 3v_t$ from $m \geq 1$ and $r > v_t$. Consequently,

$$|\mathcal{E}(\widetilde{A}^{(v_t)})| \geq n - (2v_t - r) \geq n - v_t \geq \frac{2}{3}n.$$

We have shown $|\mathcal{E}(\widetilde{A}^{(v_t)})| \geq \frac{2}{3}n$ for $v_t \leq n/2$.

For $v_t > n/2$, recall that we have shown in Appendix B.2 that the undirected graph generated with label difference $v_t$ and starting point $s_t$ equals the undirected graph generated with label difference $q = n - v_t < n/2$ and starting point $(s_t - q) \bmod n$. Since the number of edges is invariant with the starting point, the case $v_t > n/2$ has been reduced to $v_t < n/2$. This completes the proof. $\qquad\square$

**Lemma 4** *For any symmetric matrix $B = [b_{ij}] \in \mathbb{R}^{n \times n}$, if $B\mathbb{1}_n = 0_n$, then*

$$x^T B x = -\frac{1}{2} \sum_{i,j} b_{ij}(x_i - x_j)^2, \quad \forall x \in \mathbb{R}^n.$$

*Proof.* The $i$-th entry of $Bx$ is

$$[Bx]_i = \sum_j b_{ij}x_j = b_{ii}x_i + \sum_{j:j \neq i} b_{ij}x_j = \sum_j b_{ij}(x_j - x_i).$$

Hence,

$$x^T B x = \sum_{i,j} b_{ij}x_i(x_j - x_i).$$

Due to $B^T = B$, we have

$$x^T B x = \sum_{i,j} b_{ij}x_j(x_i - x_j).$$

Averaging the above equations yields the result. $\qquad\square$

**Lemma 5** *For any $n \geq 2$, it holds for $\widetilde{A}^{(v_t)}$ that*

$$\mathbb{E}[\widetilde{A}^{(v_t)}] \preceq \frac{1}{3}I + \frac{1}{3}\left(A^{(v_t)} + [A^{(v_t)}]^T\right).$$

*Proof.* If $v_t = n/2$, then $\widetilde{A}^{(v_t)} = A^{(v_t)} = [A^{(v_t)}]^T$. By Lemma 4, $x^T\left(I - A^{(v_t)}\right)x \geq 0$, i.e., $A^{(v_t)} \preceq I$. Thus,

$$\widetilde{A}^{(v_t)} = A^{(v_t)} \preceq \frac{1}{3}I + \frac{2}{3}\left(A^{(v_t)} + [A^{(v_t)}]^T\right).$$

Consider $v_t \neq n/2$. Notice that $\mathcal{E}(\widetilde{\boldsymbol{A}}^{(v_t)}) \subset \mathcal{E}(\boldsymbol{A}^{(v_t)} + [\boldsymbol{A}^{(v_t)}]^T)$ and $|\mathcal{E}(\boldsymbol{A}^{(v_t)} + [\boldsymbol{A}^{(v_t)}]^T)| \leq 2n$. By Lemma 3 and the fact that $s_t$ is from the uniform distribution over $[n]$, it holds for any $(i,j) \in \mathcal{E}(\boldsymbol{A}^{(v_t)} + [\boldsymbol{A}^{(v_t)}]^T)$ that

$$\mathbb{P}[(i,j) \in \mathcal{E}(\widetilde{\boldsymbol{A}}^{(v_t)})] \geq \frac{1}{2n}\mathbb{E}[|\mathcal{E}(\widetilde{\boldsymbol{A}}^{(v_t)})|] \geq \frac{1}{3}.$$

By the construction of $\widetilde{\boldsymbol{A}}^{(v_t)}$ in Alg. 2 and $\boldsymbol{A}^{(v_t)}$ in (2), the non-diagonal and non-zero entries are $\frac{n-1}{n}$. It follows from Lemma 4 that for any $\boldsymbol{x} \in \mathbb{R}^n$, we have

$$
\begin{aligned}
\boldsymbol{x}^T \mathbb{E}[\mathbf{I} - \widetilde{\boldsymbol{A}}^{(v_t)}]\boldsymbol{x} &= \frac{n-1}{2n}\mathbb{E}\left[\sum_{(i,j)\in\mathcal{E}(\widetilde{\boldsymbol{A}}^{(v_t)})} (x_i - x_j)^2\right] \\
&= \frac{n-1}{2n}\sum_{(i,j)\in\mathcal{E}(\boldsymbol{A}^{(v_t)}+[\boldsymbol{A}^{(v_t)}]^T)} \mathbb{P}[(i,j)\in\mathcal{E}(\widetilde{\boldsymbol{A}}^{(v_t)})](x_i-x_j)^2 \\
&\geq \frac{n-1}{6n}\sum_{(i,j)\in\mathcal{E}(\boldsymbol{A}^{(v_t)}+[\boldsymbol{A}^{(v_t)}]^T)} (x_i - x_j)^2 \\
&= \frac{1}{3}\boldsymbol{x}^T\left(2\mathbf{I} - \boldsymbol{A}^{(v_t)} - [\boldsymbol{A}^{(v_t)}]^T\right)\boldsymbol{x}.
\end{aligned}
$$

Rearranging the terms, we derive

$$\mathbb{E}\left[\widetilde{\boldsymbol{A}}^{(v_t)}\right] \preceq \frac{1}{3}\boldsymbol{I} + \frac{1}{3}\left(\boldsymbol{A}^{(v_t)} + [\boldsymbol{A}^{(v_t)}]^T\right).$$

This completes the proof. $\qquad\square$

**Proof of Theorem 4** It follows from Lemma 5 that

$$\mathbb{E}\left[\widetilde{\boldsymbol{A}}^{(v_t)}\right] \preceq \frac{1}{3}\boldsymbol{I} + \frac{1}{3}\mathbb{E}\left[\boldsymbol{A}^{(v_t)} + [\boldsymbol{A}^{(v_t)}]^T\right] = \frac{1}{3}\boldsymbol{I} + \frac{2}{3}\widetilde{\boldsymbol{W}}.$$

Consequently,

$$
\begin{aligned}
\mathbb{E}\left[\left\|\boldsymbol{\Pi}\widetilde{\boldsymbol{W}}^{(t)}\boldsymbol{x}\right\|^2\right] &= \mathbb{E}\left[\left\|\boldsymbol{\Pi}\left((1-\eta)\boldsymbol{I} + \eta\widetilde{\boldsymbol{A}}^{(v_t)}\right)\boldsymbol{x}\right\|^2\right] \\
&\leq (1-\eta)^2\|\boldsymbol{\Pi}\boldsymbol{x}\|^2 + 2\eta(1-\eta)(\boldsymbol{\Pi}\boldsymbol{x})^T\mathbb{E}\left[\widetilde{\boldsymbol{A}}^{(v_t)}\right]\boldsymbol{\Pi}\boldsymbol{x} + \eta^2\mathbb{E}\left[\left\|\widetilde{\boldsymbol{A}}^{(v_t)}\right\|_2^2\right]\|\boldsymbol{\Pi}\boldsymbol{x}\|^2.
\end{aligned}
$$

Combining the above two inequalities, we derive

$$
\begin{aligned}
\mathbb{E}\left[\left\|\boldsymbol{\Pi}\widetilde{\boldsymbol{W}}^{(t)}\boldsymbol{x}\right\|^2\right] &\leq \left((1-\eta)^2 + \frac{2}{3}\eta(1-\eta) + \frac{4}{3}\eta\rho(1-\eta) + \eta^2\right)\|\boldsymbol{\Pi}\boldsymbol{x}\|^2 \\
&= \left(1 - \frac{4}{3}\eta(1-\eta)(1-\rho)\right)\|\boldsymbol{\Pi}\boldsymbol{x}\|^2.
\end{aligned}
$$

Thus, the proof is completed. $\qquad\square$

## B.4 An alternative construction of OU-EquiDyn

Alg. 5 provides a different way to construct one-peer undirected graphs which achieve a similar consensus rate as the graphs generated by Alg. 2 but with a different structure.

We denote the matrix $\boldsymbol{A}$ generated at the $t$-th iteration of Alg. 5 by $\overline{\boldsymbol{A}}^{(v_t)} = [\overline{a}_{ij}^{(v_t)}] \in \mathbb{R}^{n\times n}$. Next, we explain the motivation of Alg. 5.

Denote $\gcd(a,b)$ as the greatest common divisor of $a$ and $b$. Let $d = \gcd(v_t, n)$ and $\tilde{v}_t = v_t/d$, $\tilde{n} = n/d$. Then, $\tilde{v}_t$ and $\tilde{n}$ are coprime.

Firstly, we divide $[n]$ into $d$ disjoint subsets:

$$\overline{C}_\ell = \{i \in [n] : \mod(i - s_t, d) = \ell\}, \ 0 \leq \ell < d.$$

**Algorithm 5:** Alternative OU-EquiDyn weight matrix generation at iteration $t$ (from nodes' perspective)

---

**Input:** $\eta \in (0,1)$; basis index $\{u_i, -u_i\}_{i=1}^{M}$ from a weight matrix $\widetilde{W} \in \mathbb{R}^{n \times n}$ of form 5;

Initialize $A = [a_{ij}] = I$;

**for** *node $i = 1$ to $n$ (in parallel)* **do**

    Pick $v_t$ from $\{u_i, -u_i\}_{i=1}^{M}$ and $s_t \in [n]$ uniformly at random using the common random seed;

    Compute $d = \gcd(v_t, n)$ and find $1 \le b \le n/d - 1$ such that $\mathrm{mod}(b \cdot v_t, n) = d$ by Euclidean algorithm;

    Set $\tilde{n} = n/d$, and $m(i) = \mathrm{mod}(\lfloor (i - s_t)/d \rfloor \cdot b, \tilde{n})$;

    **if** *$\tilde{n}$ is even or $m(i) < \tilde{n} - 1$* **then**

        **if** *$m$ is even* **then**

            $j = (i + v_t) \bmod n$;

        **end**

        **else**

            $j = (i - v_t) \bmod n$;

        **end**

        $a_{ij} = (n-1)/n$;

        $a_{ii} = 1/n$;

    **end**

**end**

**Output:** $\widetilde{W}^{(t)} = (1 - \eta)I + \eta A$

---

Clearly, $\left| \overline{C}_\ell \right| = n/d = \tilde{n}, \forall 0 \le \ell < d$.

We claim that

$$\overline{C}_\ell = \{ s_t + \ell + m v_t \bmod n : \ 0 \le m < \tilde{n} \}, \ 0 \le \ell < d. \tag{11}$$

To proof the claim, we denote the RHS of (11) by $\widetilde{C}_\ell$. Since $v_t$ can be divided evenly by $d$, for any $i \in \widetilde{C}_\ell$, it satisfies $\mathrm{mod}(i, d) = \mathrm{mod}((s_t + \ell), d)$. Combining with the fact that $\widetilde{C}_\ell \subset [n]$, we have $\widetilde{C}_\ell \subset \overline{C}_\ell$.

Then, since $\tilde{n}$ and $\tilde{v}$ are coprime, for any $0 \le m_1 < m_2 < \tilde{n}$, $\mathrm{mod}(m_1 \tilde{v}_t, \tilde{n}) \ne \mathrm{mod}(m_2 \tilde{v}_t, \tilde{n})$. Then, $\mathrm{mod}(m_1 v_t, n) \ne \mathrm{mod}(m_2 v_t, n)$. Thus, $\left| \widetilde{C}_\ell \right| = \tilde{n} = \left| \overline{C}_\ell \right|$. Combining with $\widetilde{C}_\ell \subset \overline{C}_\ell$, we have $\overline{C}_\ell = \widetilde{C}_\ell$.

The above analysis also implies that for each $i \in \overline{C}_\ell$, there exists a unique $0 \le m(i) < d$ such that $i = (s_t + \ell + m(i) v_t) \bmod n$.

By (11), a natural way to construct one-peer undirected graphs is: in each $\overline{C}_\ell$, connect $s_t + \ell$ with $s_t + \ell + v_t$, connect $s_t + \ell + 2 v_t$ with $s_t + \ell + 3 v_t, \ldots$. Equivalently, $i$ is connected with $(i + v_t) \bmod n$ if $m(i)$ is even and connected with $(i - v_t) \bmod n$ if $m(i)$ is odd.

In this way, if $\tilde{n}$ is even, every node in $C_\ell$ has a neighbor, $\forall 0 \le \ell < d$, i.e., every node in $[n]$ has a neighbor. Equivalently, a node $i \in \overline{C}_\ell$ has a neighbor if it is in the set

$$\overline{C}'_\ell = \begin{cases} C_\ell, & \text{if } \tilde{n} \text{ is even} \\ \{ (s_t + \ell + m v_t) \bmod n : \ 0 \le m < \tilde{n} - 1 \}, & \text{if } \tilde{n} \text{ is odd} \end{cases} \tag{12}$$

To give a practical way of the process described above from each node's perspective, for each node $i \in \overline{C}_\ell$, we provide a more efficient way to compute the unique $0 \le m(i) < \tilde{n}$ such that $i = (s_t + \ell + m(i) v_t) \bmod n$. Firstly, for each $i \in \overline{C}_\ell$, since $\ell = \mathrm{mod}(i - s_t, d)$, we have

$$i = (s_t + \ell + \lfloor (i - s_t)/d \rfloor \cdot d) \bmod n.$$

Then, by Bézout's theorem, since $\tilde{n}$ and $\tilde{v}_t$ are coprime, there exist integers $1 \leq b \leq \tilde{n} - 1$ and $b'$ such that $b \cdot v_t + b' \cdot n = d$. The pair $(b, b')$ can be computed by the Euclidean algorithm in $\mathcal{O}(\ln(n))$ time. Then,

$$\lfloor (i - s_t)/d \rfloor \cdot d = \lfloor (i - s_t)/d \rfloor \cdot (bv_t + b'n).$$

Define $m(i) = \mathrm{mod}(\lfloor (i - s_t)/d \rfloor \cdot b, \tilde{n})$, then, we have $(\lfloor (i - s_t)/d \rfloor \cdot d) \bmod n = m(i)v_t \bmod n$, Thus, $i = (s_t + \ell + m(i)v_t) \bmod n$.

In Alg. 5, each node $i$ computes $\tilde{n}$ and its $m(i)$ firstly. If $\tilde{n}$ is even, every node has a neighbor. If $\tilde{n}$ is odd but $m(i) < \tilde{n} - 1$, then $i \in \overline{C}'_\ell$ where $\ell = \mod(i - s_t, d)$, i.e., $i$ also has a neighbor. In this way, each node can determine whether it has a neighbor in this iteration. If node $i$ has a neighbor, as we have described above, it is connected with $(i + v_t) \bmod n$ if $m(i)$ is even and $(i - v_t) \bmod n$ otherwise.

The following lemma is used to prove Lemma 7.

**Lemma 6** *Let $d = \gcd(v_t, n)$, then*

$$\mathbb{E}\big[|\mathcal{E}(\overline{\boldsymbol{A}}^{(v_t)})|\big] = (2d) \cdot \left\lfloor \frac{n}{2d} \right\rfloor \geq \frac{2n}{3}.$$

*Proof.* Define $\overline{C}_\ell$ and $\overline{C}'_\ell$ as in (11) and (12). If $n$ can be divided by $2d$ evenly, i.e., $|\overline{C}_\ell| = \tilde{n}$ is even for any $0 \leq \ell < d$. Then, every node has a neighbor, i.e., $\left|\mathcal{E}(\overline{\boldsymbol{A}}^{(v_t)})\right| = n = (2d) \cdot \left\lfloor \frac{n}{2d} \right\rfloor$.

If $n$ cannot be divided by $2d$ evenly, by (12), in each $\overline{C}_\ell$, there is one node idle in this iteration. Then, we also have $\left|\mathcal{E}(\overline{\boldsymbol{A}}^{(v_t)})\right| = n - d = (2d) \cdot \left\lfloor \frac{n}{2d} \right\rfloor$. As $d = \gcd(v_t, n)$, we have $n = (2k + 1)d$, with $k \in \mathbb{Z}$. Since $v_t$ can be divided by $d$ evenly and $v_t \leq n - 1$, we have $d < n$. Thus, $k \geq 1$, i.e., $n \geq 3d$. Then, $\left|\mathcal{E}(\overline{\boldsymbol{A}}^{(v_t)})\right| = n - d \geq n - \frac{n}{3} = \frac{2n}{3}$.

Since the above analysis holds for arbitrary $s_t \in [n]$, the lemma is proved. $\qquad\square$

The following lemma follows by similar arguments with Lemma 5.

**Lemma 7** *For any $n \geq 2$, the output matrix $\overline{\boldsymbol{A}}^{(v_t)}$ of Algorithm 2 satisfies*

$$\mathbb{E}[\overline{\boldsymbol{A}}^{(v_t)}] \preceq \frac{1}{3}\boldsymbol{I} + \frac{1}{3}\Big(\boldsymbol{A}^{(v_t)} + [\boldsymbol{A}^{(v_t)}]^T\Big).$$

The following consensus rate for Alg. 5 is proved similarly to Theorem 4.

**Theorem 7** *Let $\widetilde{\boldsymbol{W}}$ be a U-EquiStatic matrix with consensus rate $\rho$, and $\widetilde{\boldsymbol{W}}^{(t)}$ be an OU-EquiDyn matrix generated by Alg. 5, it holds that*

$$\mathbb{E}\left[\left\|\boldsymbol{\Pi}\widetilde{\boldsymbol{W}}^{(t)}\boldsymbol{x}\right\|^2\right] \leq \big(1 - \frac{4}{3}\eta(1-\eta)(1-\rho)\big)\left\|\boldsymbol{\Pi}\boldsymbol{x}\right\|^2, \quad \forall \boldsymbol{x} \in \mathbb{R}^n.$$

## C Applying EquiTopo Matrices to Decentralized Learning

### C.1 Convergence of DSGD for strongly convex cost functions

We assume that $f_i(\boldsymbol{x})$ is $\mu$-strongly convex for any $i$, i.e., there exists a constant $\mu > 0$ such that

$$f_i(\boldsymbol{y}) \geq f_i(\boldsymbol{x}) + \langle \nabla f_i(\boldsymbol{x}), \boldsymbol{y} - \boldsymbol{x} \rangle + \frac{\mu}{2}\left\|\boldsymbol{y} - \boldsymbol{x}\right\|^2, \quad \forall \boldsymbol{x}, \boldsymbol{y} \in \mathbb{R}^d.$$

As we have tested the performance of DSGD with EquiTopo matrices for strongly convex cost functions, we attach the following convergence result of the algorithm (8). The proof follows by [12, Theorem 2] (or Appendix A.4 therein) and is omitted here.

**Theorem 8** *Consider the DSGD algorithm* (8) *utilizing the EquiTopo matrices, and $f_i$ being $\mu$-strongly convex for all $i$. Under Assumptions A.1-A.3, it holds that*

$$\frac{1}{H_T}\sum_{t=0}^{T}h^{(t)}\mathbb{E}\Big[f(\bar{\boldsymbol{x}}^{(t)})-f(\boldsymbol{x}^*)\Big]=\tilde{\mathcal{O}}\Big(\frac{\sigma^2}{nT}+\frac{\kappa\beta\sigma^2}{(1-\beta)T^2}+\frac{\kappa\beta b^2}{(1-\beta)^2T^2}+\frac{1}{1-\beta}\exp\big(-\frac{(1-\beta)T}{\kappa}\big)\Big)$$

*where $\kappa=L/\mu$, $\tilde{\mathcal{O}}(\cdot)$ hides constants and polylogarithmic factors, positive weights $h^{(t)}=\big(1-\frac{\mu\gamma}{2}\big)^t$, and $H_T=\sum_{t=0}^{T}h^{(t)}$. Furthermore,*

- $\beta=\rho$ *with D-EquiStatic $\boldsymbol{W}$ or U-EquiStatic $\widetilde{\boldsymbol{W}}$;*

- $\beta=\sqrt{(1+\rho)/2}$ *for OD-EquiDyn $\boldsymbol{W}^{(t)}$ (Alg. 1 with $\eta=1/2$), and $\beta=\sqrt{(2+\rho)/3}$ for OU-EquiDyn $\widetilde{\boldsymbol{W}}^{(t)}$ (Alg. 2 with $\eta=1/2$).*

## C.2 Transient iteration

**The computation of transient iteration.**

For nonconvex cost functions, the convergence rate of (8) is given by

$$\frac{1}{T}\sum_{t=0}^{T-1}\mathbb{E}\Big[\big\|\nabla f(\bar{\boldsymbol{x}}^{(t)})\big\|^2\Big]=\mathcal{O}\Big(\frac{\sigma}{\sqrt{nT}}+\frac{\beta^{\frac{2}{3}}\sigma^{\frac{2}{3}}}{T^{\frac{2}{3}}(1-\beta)^{\frac{1}{3}}}+\frac{\beta^{\frac{2}{3}}b^{\frac{2}{3}}}{T^{\frac{2}{3}}(1-\beta)^{\frac{2}{3}}}+\frac{\beta}{T(1-\beta)}\Big)$$

To reach the linear speedup stage, the iteration T has to be sufficiently large so that the $\sqrt{nT}$-term dominates, i.e., $\frac{\sigma}{\sqrt{nT}}\geq\frac{\beta^{\frac{2}{3}}\sigma^{\frac{2}{3}}}{T^{\frac{2}{3}}(1-\beta)^{\frac{1}{3}}}$, $\frac{\sigma}{\sqrt{nT}}\geq\frac{\beta^{\frac{2}{3}}b^{\frac{2}{3}}}{T^{\frac{2}{3}}(1-\beta)^{\frac{2}{3}}}$, and moreover, $\frac{\sigma}{\sqrt{nT}}\geq\frac{\beta}{T(1-\beta)}$. Then $T\geq\frac{\beta^4 n^3}{(1-\beta)^2\sigma^2}$, $T\geq\frac{\beta^4 b^4 n^3}{(1-\beta)^4\sigma^6}$, and $T\geq\frac{\beta^2 n}{(1-\beta)^2\sigma^2}$. Substituting $\beta$ into the inequalities, transient iterations under different networks can be computed. Similar methods can be adopted for the transient iterations of the distributed gradient tracking algorithm.

Under different network topologies, for non-convex and strongly convex cost functions, convergence results and transient iterations are shown in Table 3 and 4. The results indicate that the proposed networks are at faster rates.

Table 3: For non-convex cost functions, per-iteration communication and convergence rate comparison between DSGD over different topologies. The smaller the transient iteration complexity is, the faster the algorithm converges.

| Topology | Per-iter Comm. | Convergence Rate | Trans. Iters. |
|---|---|---|---|
| Ring | $\Theta(1)$ | $\mathcal{O}\Big(\frac{\sigma}{\sqrt{nT}}+\frac{n^{\frac{2}{3}}\sigma^{\frac{2}{3}}}{T^{\frac{2}{3}}}+\frac{n^{\frac{4}{3}}b^{\frac{2}{3}}}{T^{\frac{2}{3}}}+\frac{n^2}{T}\Big)$ | $\mathcal{O}(n^{11})$ |
| Torus | $\Theta(1)$ | $\mathcal{O}\Big(\frac{\sigma}{\sqrt{nT}}+\frac{n^{\frac{1}{3}}\sigma^{\frac{2}{3}}}{T^{\frac{2}{3}}}+\frac{n^{\frac{2}{3}}b^{\frac{2}{3}}}{T^{\frac{2}{3}}}+\frac{n}{T}\Big)$ | $\mathcal{O}(n^7)$ |
| Static Exp. | $\Theta(\ln(n))$ | $\mathcal{O}\Big(\frac{\sigma}{\sqrt{nT}}+\frac{\ln^{\frac{1}{3}}(n)\sigma^{\frac{2}{3}}}{T^{\frac{2}{3}}}+\frac{\ln^{\frac{2}{3}}(n)b^{\frac{2}{3}}}{T^{\frac{2}{3}}}+\frac{\ln(n)}{T}\Big)$ | $\mathcal{O}(n^3\ln^4(n))$ |
| O.-P. Exp. | $1$ | $\mathcal{O}\Big(\frac{\sigma}{\sqrt{nT}}+\frac{\ln^{\frac{1}{3}}(n)\sigma^{\frac{2}{3}}}{T^{\frac{2}{3}}}+\frac{\ln^{\frac{2}{3}}(n)b^{\frac{2}{3}}}{T^{\frac{2}{3}}}+\frac{\ln(n)}{T}\Big)$ | $\mathcal{O}(n^3\ln^4(n))$ |
| D(U)-EquiStatic | $\Theta(\ln(n))$ | $\mathcal{O}\Big(\frac{\sigma}{\sqrt{nT}}+\frac{\sigma^{\frac{2}{3}}}{T^{\frac{2}{3}}}+\frac{b^{\frac{2}{3}}}{T^{\frac{2}{3}}}+\frac{1}{T}\Big)$ | $\mathcal{O}(n^3)$ |
| OD (OU)-EquiDyn | $1$ | $\mathcal{O}\Big(\frac{\sigma}{\sqrt{nT}}+\frac{\sigma^{\frac{2}{3}}}{T^{\frac{2}{3}}}+\frac{b^{\frac{2}{3}}}{T^{\frac{2}{3}}}+\frac{1}{T}\Big)$ | $\mathcal{O}(n^3)$ |

## C.3 Decentralized stochastic gradient tracking algorithm

We write local variables compactly into matrix form, for instance

$$\boldsymbol{X}^{(t)}=\Big[\boldsymbol{x}_1^{(t)},\cdots,\boldsymbol{x}_n^{(t)}\Big]^T\in\mathbb{R}^{n\times d},\ \nabla\boldsymbol{F}\big(\boldsymbol{X}^{(t)}\big)=\Big[\nabla f_1\big(\boldsymbol{x}_1^{(t)}\big),\cdots,\nabla f_n\big(\boldsymbol{x}^{(t)}\big)\Big]^T\in\mathbb{R}^{n\times d}.$$

Table 4: For strongly convex cost functions, per-iteration communication and convergence rate comparison between DSGD over different topologies. The smaller the transient iteration complexity is, the faster the algorithm converges.

| Topology | Per-iter Comm. | Convergence Rate | Trans. Iters. |
|---|---|---|---|
| Ring | $\Theta(1)$ | $\tilde{\mathcal{O}}\left(\frac{\sigma^2}{nT} + \frac{\kappa n^2 \sigma^2}{T^2} + \frac{\kappa n^4 b^2}{T^2}\right)$ | $\tilde{\mathcal{O}}(\kappa n^5)$ |
| Torus | $\Theta(1)$ | $\tilde{\mathcal{O}}\left(\frac{\sigma^2}{nT} + \frac{\kappa n \sigma^2}{T^2} + \frac{\kappa n^2 b^2}{T^2}\right)$ | $\tilde{\mathcal{O}}(\kappa n^3)$ |
| Static Exp. | $\Theta(\ln(n))$ | $\tilde{\mathcal{O}}\left(\frac{\sigma^2}{nT} + \frac{\kappa \ln(n)\sigma^2}{T^2} + \frac{\kappa \ln^2(n) b^2}{T^2}\right)$ | $\tilde{\mathcal{O}}(\kappa n \ln^2(n))$ |
| O.-P. Exp. | 1 | $\tilde{\mathcal{O}}\left(\frac{\sigma^2}{nT} + \frac{\kappa \ln(n)\sigma^2}{T^2} + \frac{\kappa \ln^2(n) b^2}{T^2}\right)$ | $\tilde{\mathcal{O}}(\kappa n \ln^2(n))$ |
| D(U)-EquiStatic | $\Theta(\ln(n))$ | $\tilde{\mathcal{O}}\left(\frac{\sigma^2}{nT} + \frac{\kappa \sigma^2}{T^2} + \frac{\kappa b^2}{T^2}\right)$ | $\tilde{\mathcal{O}}(\kappa n)$ |
| OD (OU)-EquiDyn | 1 | $\tilde{\mathcal{O}}\left(\frac{\sigma^2}{nT} + \frac{\kappa \sigma^2}{T^2} + \frac{\kappa b^2}{T^2}\right)$ | $\tilde{\mathcal{O}}(\kappa n)$ |

The matrices $\boldsymbol{Y}^{(t)}, \boldsymbol{G}^{(t)} \in \mathbb{R}^{n \times d}$ are defined analogously. We also denote $\nabla \boldsymbol{F}(\boldsymbol{X}^{(-1)}) = \boldsymbol{G}^{(-1)} = \boldsymbol{0}$ for notational simplicity.

Clearly, the DSGT algorithm can be simplified as

$$\begin{pmatrix} \boldsymbol{X}^{(t+1)} \\ \boldsymbol{Y}^{(t+1)} \end{pmatrix} = \begin{pmatrix} \boldsymbol{W}^{(t)} & -\gamma \boldsymbol{W}^{(t)} \\ \boldsymbol{0} & \boldsymbol{W}^{(t)} \end{pmatrix} \begin{pmatrix} \boldsymbol{X}^{(t)} \\ \boldsymbol{Y}^{(t)} \end{pmatrix} + \begin{pmatrix} \boldsymbol{0} \\ \boldsymbol{G}^{(t+1)} - \boldsymbol{G}^{(t)} \end{pmatrix}. \tag{13}$$

For simplicity, we define

$$\boldsymbol{W}^{(j:k)} = \boldsymbol{W}^{(k)} \cdots \boldsymbol{W}^{(j)}, \ \ \forall k \geq j \geq 0,$$

and moreover, $\boldsymbol{W}^{(j:k)} = \boldsymbol{I}$ for $j > k$.

Notice that

$$\begin{pmatrix} \boldsymbol{W}^{(k)} & -\gamma \boldsymbol{W}^{(k)} \\ \boldsymbol{0} & \boldsymbol{W}^{(k)} \end{pmatrix} \cdots \begin{pmatrix} \boldsymbol{W}^{(j)} & -\gamma \boldsymbol{W}^{(j)} \\ \boldsymbol{0} & \boldsymbol{W}^{(j)} \end{pmatrix} = \begin{pmatrix} \boldsymbol{W}^{(j:k)} & -\gamma (k-j+1) \boldsymbol{W}^{(j:k)} \\ \boldsymbol{0} & \boldsymbol{W}^{(j:k)} \end{pmatrix}.$$

Consequently, it holds for all $t \geq 1$ that

$$\boldsymbol{X}^{(t)} = \boldsymbol{W}^{(0:(t-1))} \boldsymbol{X}^{(0)} - \gamma \sum_{j=0}^{t-1} (t-j) \boldsymbol{W}^{(j:(t-1))} \left( \boldsymbol{G}^{(j)} - \boldsymbol{G}^{(j-1)} \right). \tag{14}$$

Moreover, we have two inequalities as follows.

**Lemma 8** *For any $t \geq 1$ and $\beta \in (0,1)$, we have*

$$t^2 \beta^{2(t-1)} \leq \frac{c_1}{(1-\beta)^2} \beta^{t-1}, \quad t^2 \beta^{t-1} \leq \frac{c_2}{(1-\beta)^2} \left(\frac{1+\beta}{2}\right)^{t-1},$$

*where $c_1 = 4$, $c_2 = 16$.*

*Proof.* Define $r(x) = x^2 \beta^{x-1}$, where $x \geq 1$. Then, for the first inequality, it suffices to show that $r(x) \leq c_1/(1-\beta)^2$ for $x \geq 1$. Due to $r'(x) = x\beta^{x-1}(2 + x \ln \beta)$, $r(x)$ attains its maximum at $x_0 = \max\left\{1, -\frac{2}{\ln \beta}\right\}$.

If $-\frac{2}{\ln \beta} > 1$, by combining with the fact that $\ln \beta \leq \beta - 1 < 0$ for $\beta \in (0,1)$, we have

$$x^2 \beta^{x-1} \leq r\left(-\frac{2}{\ln \beta}\right) = \frac{4}{(\ln \beta)^2} \beta^{-\frac{2}{\ln \beta} - 1} \leq \frac{4}{(\ln \beta)^2} \leq \frac{4}{(1-\beta)^2}.$$

If $-\frac{2}{\ln \beta} \le 1$, then

$$x^2 \beta^{x-1} = r(1) = 1 \le \frac{4}{(1-\beta)^2}.$$

The second inequality follows by similar arguments and the fact that $\sqrt{\beta} \le \frac{1+\beta}{2}$. The proof is completed. $\qquad\square$

The following lemma is a generalization of Cauchy-Schwartz inequality. Its proof follows by using $\|\boldsymbol{A} + \boldsymbol{B}\|_F^2 \le \frac{1}{\alpha} \|\boldsymbol{A}\|_F^2 + \frac{1}{1-\alpha} \|\boldsymbol{B}\|_F^2$ ($\alpha \in (0,1)$) repeatedly.

**Lemma 9** *Consider a sequence of matrices $\{\boldsymbol{B}_i\}_{i=1}^m$. If $\alpha_1, \alpha_2, \cdots, \alpha_m > 0$ and $\sum_{i=1}^m \alpha_i \le 1$, then*

$$\left\| \sum_{i=1}^m \boldsymbol{B}_i \right\|_F^2 \le \sum_{i=1}^m \frac{1}{\alpha_i} \|\boldsymbol{B}_i\|_F^2.$$

We define a potential function as

$$\Phi^{(t)} = \frac{4\beta^t}{(1-\beta)^2} \mathbb{E}\left[\left\|\boldsymbol{\Pi}\boldsymbol{X}^{(0)}\right\|_F^2\right] + \frac{16c_1 n\gamma^2}{(1-\beta)^4}\sigma^2$$
$$+ \frac{4c_2\gamma^2}{(1-\beta)^4} \sum_{j=0}^{t-1} \left(\frac{1+\beta}{2}\right)^{t-j-1} \mathbb{E}\left[\left\|\nabla\boldsymbol{F}\left(\boldsymbol{X}^{(j)}\right) - \nabla\boldsymbol{F}\left(\boldsymbol{X}^{(j-1)}\right)\right\|_F^2\right], \quad \forall t \ge 1, \tag{15}$$

and moreover,

$$\Phi^{(0)} = \frac{4}{(1-\beta)^2} \left\|\boldsymbol{\Pi}\boldsymbol{X}^{(0)}\right\|_F^2 + \frac{16c_1 n\gamma^2}{(1-\beta)^4}\sigma^2. \tag{16}$$

The following Lemma 10 and Lemma 11 are used to prove Lemma 12. Theorem 6 follows by combining Lemma 12 with the descent lemma (Lemma 14).

**Lemma 10** *Consider the DSGT (9). Let Assumptions A.1 and A.2 hold. If $\left\{\boldsymbol{W}^{(t)}\right\}_{t\ge 0}$ have convergence rate $\beta$, i.e., $\mathbb{E}\left[\left\|\boldsymbol{\Pi}\boldsymbol{W}^{(t)}\boldsymbol{y}\right\|^2\right] \le \beta^2 \|\boldsymbol{\Pi}\boldsymbol{y}\|^2$ for any $\boldsymbol{y} \in \mathbb{R}^n$, then*

$$\mathbb{E}\left[\left\|\boldsymbol{\Pi}\boldsymbol{X}^{(t)}\right\|_F^2\right] \le \frac{1-\beta}{2}\Phi^{(t)}, \quad \forall t \ge 0.$$

*Proof.* The case $t = 0$ follows by definition directly.

For $t \ge 1$, we define

$$\boldsymbol{Q}^{(t,1)} = \boldsymbol{W}^{(0:(t-1))}\boldsymbol{X}^{(0)} - \gamma \sum_{j=0}^{t-1}(t-j)\,\boldsymbol{W}^{(j:(t-1))}\left(\nabla\boldsymbol{F}\left(\boldsymbol{X}^{(j)}\right) - \nabla\boldsymbol{F}\left(\boldsymbol{X}^{(j-1)}\right)\right),$$

and

$$\boldsymbol{Q}^{(t,2)} = -\gamma \sum_{j=0}^{t-1}(t-j)\,\boldsymbol{W}^{(j:(t-1))}\left(\boldsymbol{G}^{(j)} - \nabla\boldsymbol{F}\left(\boldsymbol{X}^{(j)}\right) - \boldsymbol{G}^{(j-1)} + \nabla\boldsymbol{F}\left(\boldsymbol{X}^{(j-1)}\right)\right).$$

Recalling (14) gives

$$\boldsymbol{X}^{(t)} = \boldsymbol{Q}^{(t,1)} + \boldsymbol{Q}^{(t,2)}.$$

Then

$$\mathbb{E}\left[\left\|\boldsymbol{\Pi}\boldsymbol{X}^{(t)}\right\|_F^2\right] \le 2\mathbb{E}\left[\left\|\boldsymbol{\Pi}\boldsymbol{Q}^{(t,1)}\right\|_F^2\right] + 2\mathbb{E}\left[\left\|\boldsymbol{\Pi}\boldsymbol{Q}^{(t,2)}\right\|_F^2\right].$$

Rearranging $\boldsymbol{Q}^{(t,2)}$ yields

$$
\begin{aligned}
\boldsymbol{Q}^{(t,2)} = &- \gamma \boldsymbol{W}^{(t-1)} \left( \boldsymbol{G}^{(t-1)} - \nabla \boldsymbol{F}(\boldsymbol{X}^{(t-1)}) \right) \\
&- \gamma \sum_{j=0}^{t-2} \left( (t-j) \, \boldsymbol{W}^{(j:(t-1))} - (t-j-1) \, \boldsymbol{W}^{((j+1):(t-1))} \right) \left( \boldsymbol{G}^{(j)} - \nabla \boldsymbol{F}(\boldsymbol{X}^{(j)}) \right).
\end{aligned}
$$

By Assumption A.2 and the assumption on consensus rate, we have

$$
\begin{aligned}
&\mathbb{E} \left[ \left\| \boldsymbol{\Pi} \boldsymbol{Q}^{(t,2)} \right\|_{\mathrm{F}}^2 \right] \\
=& \gamma^2 \mathbb{E} \left[ \left\| \boldsymbol{\Pi} \boldsymbol{W}^{(t-1)} \left( \boldsymbol{G}^{(t-1)} - \nabla \boldsymbol{F}(\boldsymbol{X}^{(t-1)}) \right) \right\|_{\mathrm{F}}^2 \right] \\
&+ \gamma^2 \sum_{j=0}^{t-2} \mathbb{E} \left[ \left\| \left( (t-j) \, \boldsymbol{\Pi} \boldsymbol{W}^{(j:(t-1))} - (t-j-1) \, \boldsymbol{\Pi} \boldsymbol{W}^{((j+1):(t-1))} \right) \left( \boldsymbol{G}^{(j)} - \nabla \boldsymbol{F}(\boldsymbol{X}^{(j)}) \right) \right\|_{\mathrm{F}}^2 \right] \\
\leq& \gamma^2 \beta^2 \mathbb{E} \left[ \left\| \boldsymbol{G}^{(t-1)} - \nabla \boldsymbol{F}(\boldsymbol{X}^{(t-1)}) \right\|_{\mathrm{F}}^2 \right] + \gamma^2 \sum_{j=0}^{t-2} 4 \, (t-j)^2 \, \beta^{2(t-j-1)} \mathbb{E} \left[ \left\| \boldsymbol{G}^{(j)} - \nabla \boldsymbol{F}(\boldsymbol{X}^{(j)}) \right\|_{\mathrm{F}}^2 \right] \\
\leq& 4 n \gamma^2 \sum_{j=1}^{t} j^2 \beta^{2(j-1)} \sigma^2.
\end{aligned}
$$

By Lemma 8, we have

$$
n \gamma^2 \sum_{j=1}^{t} j^2 \beta^{2(j-1)} \sigma^2 \leq \frac{n c_1 \gamma^2}{(1-\beta)^2} \sum_{j=1}^{t} \beta^{j-1} \sigma^2 \leq \frac{n c_1 \gamma^2}{(1-\beta)^3} \sigma^2.
$$

As a result,

$$
\mathbb{E} \left[ \left\| \boldsymbol{\Pi} \boldsymbol{Q}^{(t,2)} \right\|_{\mathrm{F}}^2 \right] \leq \frac{4 n c_1 \gamma^2}{(1-\beta)^3} \sigma^2.
$$

Moreover,

$$
\begin{aligned}
&\mathbb{E} \left[ \left\| \boldsymbol{\Pi} \boldsymbol{Q}^{(t,1)} \right\|_{\mathrm{F}}^2 \right] \\
\leq& \frac{1}{(1-\beta)\,\beta^t} \mathbb{E} \left[ \left\| \boldsymbol{\Pi} \boldsymbol{W}^{(0:(t-1))} \boldsymbol{X}^{(0)} \right\|_{\mathrm{F}}^2 \right] \\
&+ \sum_{j=0}^{t-1} \frac{\gamma^2 \, (t-j)^2}{(1-\beta)\,\beta^{t-j-1}} \mathbb{E} \left[ \left\| \boldsymbol{\Pi} \boldsymbol{W}^{(j:(t-1))} \left( \nabla \boldsymbol{F}(\boldsymbol{X}^{(j)}) - \nabla \boldsymbol{F}(\boldsymbol{X}^{(j-1)}) \right) \right\|_{\mathrm{F}}^2 \right] \\
\leq& \frac{\beta^t}{(1-\beta)} \mathbb{E} \left[ \left\| \boldsymbol{\Pi} \boldsymbol{X}^{(0)} \right\|_{\mathrm{F}}^2 \right] + \frac{\gamma^2}{1-\beta} \sum_{j=0}^{t-1} (t-j)^2 \beta^{t-j-1} \mathbb{E} \left[ \left\| \nabla \boldsymbol{F}(\boldsymbol{X}^{(j)}) - \nabla \boldsymbol{F}(\boldsymbol{X}^{(j-1)}) \right\|_{\mathrm{F}}^2 \right] \\
\leq& \frac{\beta^t}{1-\beta} \mathbb{E} \left[ \left\| \boldsymbol{\Pi} \boldsymbol{X}^{(0)} \right\|_{\mathrm{F}}^2 \right] + \frac{c_2 \gamma^2}{(1-\beta)^3} \sum_{j=0}^{t-1} \left( \frac{1+\beta}{2} \right)^{t-j-1} \mathbb{E} \left[ \left\| \nabla \boldsymbol{F}(\boldsymbol{X}^{(j)}) - \nabla \boldsymbol{F}(\boldsymbol{X}^{(j-1)}) \right\|_{\mathrm{F}}^2 \right],
\end{aligned}
$$

where the first inequality follows by Lemma 9 and the fact that $(1-\beta) \sum_{j=0}^{t} \beta^j < 1$, the second inequality is by the assumption on consensus rate, and the third inequality is by Lemma 8.

Therefore, the conclusion holds by the definition of $\Phi^{(t)}$. $\qquad\square$

**Lemma 11** *Consider the DSGT* (9). *Let Assumptions A.1 and A.2 hold. If $\gamma \leq \frac{1}{L}$, it holds for $t \geq 0$ that*

$$\mathbb{E}\left[\left\|\nabla \boldsymbol{F}(\boldsymbol{X}^{(t+1)}) - \nabla \boldsymbol{F}(\boldsymbol{X}^{(t)})\right\|_{\mathrm{F}}^2\right]$$

$$\leq 6n\gamma^2 L^2 \mathbb{E}\left[\left\|\nabla f(\bar{\boldsymbol{x}}^{(t)})\right\|^2\right] + 9L^2 \mathbb{E}\left[\left\|\boldsymbol{\Pi}\boldsymbol{X}^{(t)}\right\|_{\mathrm{F}}^2\right] + 3L^2 \mathbb{E}\left[\left\|\boldsymbol{\Pi}\boldsymbol{X}^{(t+1)}\right\|_{\mathrm{F}}^2\right] + 3\gamma^2 L^2 \sigma^2.$$

*Proof.* Clearly,

$$\mathbb{E}\left[\left\|\nabla \boldsymbol{F}(\boldsymbol{X}^{(t+1)}) - \nabla \boldsymbol{F}(\boldsymbol{X}^{(t)})\right\|_{\mathrm{F}}^2\right] \leq 3\mathbb{E}\left[\left\|\nabla \boldsymbol{F}(\bar{\boldsymbol{X}}^{(t+1)}) - \nabla \boldsymbol{F}(\bar{\boldsymbol{X}}^{(t)})\right\|_{\mathrm{F}}^2\right]$$

$$+ 3\mathbb{E}\left[\left\|\nabla \boldsymbol{F}(\boldsymbol{X}^{(t+1)}) - \nabla \boldsymbol{F}(\bar{\boldsymbol{X}}^{(t+1)})\right\|_{\mathrm{F}}^2\right] + 3\mathbb{E}\left[\left\|\nabla \boldsymbol{F}(\boldsymbol{X}^{(t)}) - \nabla \boldsymbol{F}(\bar{\boldsymbol{X}}^{(t)})\right\|_{\mathrm{F}}^2\right].$$

It follows from Assumption A.1 that

$$\mathbb{E}\left[\left\|\nabla \boldsymbol{F}(\boldsymbol{X}^{(t+1)}) - \nabla \boldsymbol{F}(\boldsymbol{X}^{(t)})\right\|_{\mathrm{F}}^2\right]$$

$$\leq 3L^2 \left(\mathbb{E}\left[\left\|\bar{\boldsymbol{X}}^{(t+1)} - \bar{\boldsymbol{X}}^{(t)}\right\|_{\mathrm{F}}^2\right] + \mathbb{E}\left[\left\|\boldsymbol{\Pi}\boldsymbol{X}^{(t+1)}\right\|_{\mathrm{F}}^2\right] + \mathbb{E}\left[\left\|\boldsymbol{\Pi}\boldsymbol{X}^{(t)}\right\|_{\mathrm{F}}^2\right]\right). \tag{17}$$

Notice that by induction, $\sum_{i=1}^n \boldsymbol{y}^{(t)} = \sum_{i=1}^n \boldsymbol{g}^{(t)}$. Recalling (9) gives

$$\bar{\boldsymbol{x}}^{(t+1)} - \bar{\boldsymbol{x}}^{(t)} = \frac{\gamma}{n} \sum_{i=1}^n \boldsymbol{y}^{(t)} = \frac{\gamma}{n} \sum_{i=1}^n \boldsymbol{g}^{(t)}$$

$$= \frac{\gamma}{n} \sum_{i=1}^n \left[\nabla f_i(\bar{\boldsymbol{x}}^{(t)}) + \left(\nabla f_i(\boldsymbol{x}_i^{(t)}) - \nabla f_i(\bar{\boldsymbol{x}}^{(t)})\right) + \left(\boldsymbol{g}_i^{(t)} - \nabla f_i(\boldsymbol{x}_i^{(t)})\right)\right] \tag{18}$$

$$= \gamma \nabla f(\bar{\boldsymbol{x}}^{(t)}) + \frac{\gamma}{n} \sum_{i=1}^n \left[\left(\nabla f_i(\boldsymbol{x}_i^{(t)}) - \nabla f_i(\bar{\boldsymbol{x}}^{(t)})\right) + \left(\boldsymbol{g}_i^{(t)} - \nabla f_i(\boldsymbol{x}_i^{(t)})\right)\right].$$

By Assumptions A.1 and A.2, we derive

$$\mathbb{E}\left[\left\|\bar{\boldsymbol{x}}^{(t+1)} - \bar{\boldsymbol{x}}^{(t)}\right\|^2\right]$$

$$= \gamma^2 \mathbb{E}\left[\left\|\nabla f(\bar{\boldsymbol{x}}^{(t)}) + \frac{1}{n} \sum_{i=1}^n \left(\nabla f_i(\boldsymbol{x}_i^{(t)}) - \nabla f_i(\bar{\boldsymbol{x}}^{(t)})\right)\right\|^2\right] + \frac{\gamma^2 \sigma^2}{n} \tag{19}$$

$$\leq 2\gamma^2 \mathbb{E}\left[\left\|\nabla f(\bar{\boldsymbol{x}}^{(t)})\right\|^2\right] + \frac{2\gamma^2 L^2}{n} \mathbb{E}\left[\left\|\boldsymbol{\Pi}\boldsymbol{X}^{(t)}\right\|_{\mathrm{F}}^2\right] + \frac{\gamma^2 \sigma^2}{n}.$$

Due to $\gamma \leq \frac{1}{L}$, we have

$$\mathbb{E}\left[\left\|\bar{\boldsymbol{x}}^{(t+1)} - \bar{\boldsymbol{x}}^{(t)}\right\|^2\right] \leq 2\gamma^2 \mathbb{E}\left[\left\|\nabla f(\bar{\boldsymbol{x}}^{(t)})\right\|^2\right] + \frac{2}{n} \mathbb{E}\left[\left\|\boldsymbol{\Pi}\boldsymbol{X}^{(t)}\right\|_{\mathrm{F}}^2\right] + \frac{\gamma^2 \sigma^2}{n}. \tag{20}$$

Then the conclusion follows (17), (20) and $\left\|\bar{\boldsymbol{X}}^{(t+1)} - \bar{\boldsymbol{X}}^{(t)}\right\|_{\mathrm{F}}^2 = n \left\|\bar{\boldsymbol{x}}^{(t+1)} - \bar{\boldsymbol{x}}^{(t)}\right\|^2$. □

**Lemma 12** *Consider the DSGT* (9). *Let Assumptions A.1 and A.2 hold. Suppose that $\left\{\boldsymbol{W}^{(t)}\right\}_{t\geq 0}$ have consensus rate $\beta$ and $\boldsymbol{y} \in \mathbb{R}^n$. If $\frac{48c_2\gamma^2 L^2}{(1-\beta)^4} \leq \frac{1}{2}$, then*

$$\sum_{t=0}^T \mathbb{E}\left[\left\|\boldsymbol{\Pi}\boldsymbol{X}^{(t)}\right\|_{\mathrm{F}}^2\right] \leq 2\Phi^{(0)} + \frac{48c_2 n\gamma^4 L^2}{(1-\beta)^4} \sum_{t=1}^T \mathbb{E}\left[\left\|\nabla f(\bar{\boldsymbol{x}}^{(t-1)})\right\|^2\right] + \frac{8c_2\gamma^2}{(1-\beta)^4} \left\|\nabla \boldsymbol{F}(\boldsymbol{X}^{(0)})\right\|_{\mathrm{F}}^2$$

$$+ \frac{24c_2\gamma^4 L^2}{(1-\beta)^4} (T+1)\sigma^2 + \frac{16c_1 n\gamma^2}{(1-\beta)^3} (T+1)\sigma^2.$$

*Proof.* By the definition of $\Phi^{(t)}$ in (15), we have that for $t \geq 0$,

$$\Phi^{(t+1)} \leq \left(\frac{1+\beta}{2}\right)\Phi^{(t)} + \frac{4c_2\gamma^2}{(1-\beta)^4}\mathbb{E}\left[\left\|\nabla F(X^{(t)}) - \nabla F(X^{(t-1)})\right\|_{\mathrm{F}}^2\right] + \frac{8c_1 n\gamma^2}{(1-\beta)^3}\sigma^2. \quad (21)$$

Then, for $t \geq 0$, by Lemma 10 and (21), we have

$$\mathbb{E}\left[\left\|\Pi X^{(t)}\right\|_{\mathrm{F}}^2\right] \leq \frac{1-\beta}{2}\Phi^{(t)}$$

$$\leq \Phi^{(t)} - \Phi^{(t+1)} + \frac{4c_2\gamma^2}{(1-\beta)^4}\mathbb{E}\left[\left\|\nabla F(X^{(t)}) - \nabla F(X^{(t-1)})\right\|_{\mathrm{F}}^2\right] + \frac{8c_1 n\gamma^2}{(1-\beta)^3}\sigma^2.$$

For $t \geq 1$, by Lemma 11, we derive

$$\begin{aligned}
\mathbb{E}\left[\left\|\Pi X^{(t)}\right\|_{\mathrm{F}}^2\right] &\leq \Phi^{(t)} - \Phi^{(t+1)} + \frac{24c_2 n\gamma^4 L^2}{(1-\beta)^4}\mathbb{E}\left[\left\|\nabla f(\bar{x}^{(t-1)})\right\|^2\right] \\
&\quad + \frac{36c_2\gamma^2 L^2}{(1-\beta)^4}\mathbb{E}\left[\left\|\Pi X^{(t-1)}\right\|_{\mathrm{F}}^2\right] + \frac{12c_2\gamma^2 L^2}{(1-\beta)^4}\mathbb{E}\left[\left\|\Pi X^{(t)}\right\|_{\mathrm{F}}^2\right] \\
&\quad + \frac{12c_2\gamma^4 L^2}{(1-\beta)^4}\sigma^2 + \frac{8c_1 n\gamma^2}{(1-\beta)^3}\sigma^2.
\end{aligned} \quad (22)$$

It follows by the definition of $\Phi^{(t)}$ that

$$\Phi^{(1)} \leq \left(\frac{1+\beta}{2}\right)\Phi^{(0)} + \frac{4c_2\gamma^2}{(1-\beta)^4}\left\|\nabla F(X^{(0)})\right\|_{\mathrm{F}}^2 + \frac{8c_1 n\gamma^2}{(1-\beta)^3}\sigma^2.$$

By Lemma 10, we obtain

$$\left\|\Pi X^{(0)}\right\|_{\mathrm{F}}^2 \leq \frac{1-\beta}{2}\Phi^{(0)} \leq \Phi^{(0)} - \Phi^{(1)} + \frac{4c_2\gamma^2}{(1-\beta)^4}\left\|\nabla F(X^{(0)})\right\|_{\mathrm{F}}^2 + \frac{8c_1 n\gamma^2}{(1-\beta)^3}\sigma^2. \quad (23)$$

Taking sum on both sides of (22) and noting that $\frac{48c_2\gamma^2 L^2}{(1-\beta)^4} \leq \frac{1}{2}$, $\Phi^{(T+1)} \geq 0$, the lemma is proved. $\square$

Lemma 13 is standard in the analysis of gradient tracking methods. We attach its proof for completeness.

**Lemma 13** *Consider the DSGT (9). Let Assumptions A.1 and A.2 hold. If $\gamma \leq \frac{1}{4L}$, then*

$$\mathbb{E}\left[f(\bar{x}^{(t+1)})\right] \leq \mathbb{E}\left[f(\bar{x}^{(t)})\right] - \frac{\gamma}{4}\mathbb{E}\left[\left\|\nabla f(\bar{x}^{(t)})\right\|^2\right] + \frac{\gamma L^2}{n}\mathbb{E}\left[\left\|\Pi X^{(t)}\right\|_{\mathrm{F}}^2\right] + \frac{\gamma^2 L}{2n}\sigma^2.$$

*Proof.* By Assumption A.1, we have

$$\mathbb{E}\left[f(\bar{x}^{(t+1)})\right] \leq \mathbb{E}\left[f(\bar{x}^{(t)})\right] - \mathbb{E}\left[\left\langle\nabla f(\bar{x}^{(t)}), \bar{x}^{(t+1)} - \bar{x}^{(t)}\right\rangle\right] + \frac{L}{2}\mathbb{E}\left[\left\|\bar{x}^{(t+1)} - \bar{x}^{(t)}\right\|^2\right].$$

It follows from (9) that

$$\begin{aligned}
&\mathbb{E}\left[\left\langle\nabla f\left(\bar{x}^{(t)}\right), \bar{x}^{(t+1)} - \bar{x}^{(t)}\right\rangle\right] \\
&= \gamma\mathbb{E}\left[\left\langle\nabla f\left(\bar{x}^{(t)}\right), \nabla f(\bar{x}^{(t)}) + \frac{1}{n}\sum_{i=1}^n\left(\nabla f_i(x_i^{(t)}) - \nabla f_i(\bar{x}^{(t)})\right)\right\rangle\right] \\
&\geq \gamma\mathbb{E}\left[\left\|\nabla f(\bar{x}^{(t)})\right\|^2\right] - \frac{\gamma}{2}\mathbb{E}\left[\left\|\nabla f(\bar{x}^{(t)})\right\|^2\right] - \frac{\gamma}{2n}\sum_{i=1}^n\mathbb{E}\left[\left\|\nabla f_i(x_i^{(t)}) - \nabla f_i(\bar{x}^{(t)})\right\|\right]^2 \\
&\geq \frac{\gamma}{2}\mathbb{E}\left[\left\|\nabla f(\bar{x}^{(t)})\right\|^2\right] - \frac{\gamma L^2}{2n}\mathbb{E}\left[\left\|\Pi X^{(t)}\right\|_{\mathrm{F}}\right]^2,
\end{aligned}$$

where the first equality is by Assumption A.2 and (18); the second inequality is by Assumption A.1.

Recalling (19) yields

$$\mathbb{E}\left[f\big(\bar{\boldsymbol{x}}^{(t+1)}\big)\right]$$

$$\leq \mathbb{E}\left[f\big(\bar{\boldsymbol{x}}^{(t)}\big)\right] - \frac{\gamma}{2}\left(1 - 2\gamma L\right)\mathbb{E}\left[\left\|\nabla f\big(\bar{\boldsymbol{x}}^{(t)}\big)\right\|^2\right] + \frac{\gamma L^2}{2n}\left(1 + 2\gamma L\right)\mathbb{E}\left[\left\|\boldsymbol{\Pi}\boldsymbol{X}^{(t)}\right\|_{\mathrm{F}}^2\right] + \frac{\gamma^2\sigma^2 L}{2n}$$

$$\leq \mathbb{E}\left[f\big(\bar{\boldsymbol{x}}^{(t)}\big)\right] - \frac{\gamma}{4}\mathbb{E}\left[\left\|\nabla f\big(\bar{\boldsymbol{x}}^{(t)}\big)\right\|^2\right] + \frac{\gamma L^2}{n}\mathbb{E}\left[\left\|\boldsymbol{\Pi}\boldsymbol{X}^{(t)}\right\|_{\mathrm{F}}^2\right] + \frac{\gamma^2\sigma^2 L}{2n}.$$

The lemma is proved. □

Referring to Lemma 26 of [11], we have the following result.

**Lemma 14** *Let* $A, B, C, T$ *and* $\alpha$ *be positive constants. Define*

$$g(\gamma) = \frac{A}{\gamma T} + B\gamma + C\gamma^2.$$

*Then*

$$\inf_{\gamma\in(0,\alpha]} g\left(\gamma\right) \leq 2\left(\frac{AB}{T}\right)^{\frac{1}{2}} + 2C^{\frac{1}{3}}\left(\frac{A}{T}\right)^{\frac{2}{3}} + \frac{A}{\alpha T}.$$

**Proof of Theorem 6** Define $f^* = \inf_x f(x)$, $F_0 = f\left(\bar{\boldsymbol{x}}^{(0)}\right) - f^*$, $C_0 = \left\|\boldsymbol{\Pi}\boldsymbol{X}^{(0)}\right\|_{\mathrm{F}}^2$ and $D_0 = \sum_{i=1}^{n}\left\|\nabla f_i\left(\boldsymbol{x}_i^{(0)}\right)\right\|^2$. It suffices to show that

$$\frac{1}{T+1}\sum_{t=0}^{T}\mathbb{E}\left[\left\|\nabla f\left(\bar{\boldsymbol{x}}^{(t)}\right)\right\|_{\mathrm{F}}^2\right]$$

$$\leq \mathcal{O}\left(\sqrt{\frac{F_0 L\sigma^2}{nT}} + \frac{1}{1-\beta}\left(\frac{F_0 L\sigma}{T}\right)^{\frac{2}{3}} + \frac{F_0}{(1-\beta)^2 T} + \frac{L^2 C_0}{(1-\beta)^2 nT} + \frac{D_0}{nT}\right). \tag{24}$$

Let $\gamma \leq \frac{(1-\beta)^2}{50L}$ to satisfy the conditions in Lemmas 12 and 13. Then, we have

$$\frac{1}{T+1}\sum_{t=0}^{T}\mathbb{E}\left[\left\|\nabla f(\bar{\boldsymbol{x}}^{(t)})\right\|_{\mathrm{F}}^2\right]$$

$$\leq \frac{4}{\gamma\left(T+1\right)}\left(f\big(\bar{\boldsymbol{x}}^{(0)}\big) - f^*\right) + \frac{4L^2}{n\left(T+1\right)}\sum_{t=0}^{T}\mathbb{E}\left[\left\|\boldsymbol{\Pi}\boldsymbol{X}^{(0)}\right\|_{\mathrm{F}}^2\right] + \frac{2\gamma L}{n}\sigma^2$$

$$\leq \frac{4}{\gamma T}\left(f\big(\bar{\boldsymbol{x}}^{(0)}\big) - f^*\right) + \frac{2\gamma L}{n}\sigma^2 + \frac{8L^2}{nT}\Phi^{(0)} + \frac{32c_2\gamma^2 L^2}{(1-\beta)^4 nT}\left\|\nabla\boldsymbol{F}\big(\boldsymbol{X}^{(0)}\big)\right\|_{\mathrm{F}}^2$$

$$+ \frac{192c_2\gamma^4 L^4}{(1-\beta)^4\left(T+1\right)}\sum_{t=1}^{T}\mathbb{E}\left[\left\|\nabla f(\bar{\boldsymbol{x}}^{(t-1)})\right\|^2\right] + \frac{96c_2\gamma^4 L^4}{(1-\beta)^4 n}\sigma^2 + \frac{64c_1\gamma^2 L^2}{(1-\beta)^3}\sigma^2.$$

If $\gamma \leq \frac{1-\beta}{10L}$, then $\frac{192c_2\gamma^4 L^4}{(1-\beta)^4} \leq \frac{1}{2}$. Then, we have

$$\frac{1}{T+1}\sum_{t=0}^{T}\mathbb{E}\left[\left\|\nabla f(\bar{\boldsymbol{x}}^{(t)})\right\|_{\mathrm{F}}^2\right] \leq \frac{8}{\gamma T}\left(f\big(\bar{\boldsymbol{x}}^{(0)}\big) - f^*\right) + \frac{4\gamma L}{n}\sigma^2 + \frac{192c_2\gamma^4 L^4}{(1-\beta)^4 n}\sigma^2$$

$$+ \frac{16L^2}{nT}\Phi^{(0)} + \frac{64c_2\gamma^2 L^2}{(1-\beta)^4 nT}\left\|\nabla\boldsymbol{F}\big(\boldsymbol{X}^{(0)}\big)\right\|_{\mathrm{F}}^2 + \frac{128c_1\gamma^2 L^2}{(1-\beta)^3}\sigma^2. \tag{25}$$

By (16) and $c_1 = 4$, $c_2 = 16$ defined in Lemma 8, if $\gamma \leq \frac{(1-\beta)^2}{50L}$ and $T \geq \frac{1}{1-\beta}$, we have

$$
\begin{aligned}
\frac{1}{T+1} &\sum_{t=0}^{T} \mathbb{E}\left[\left\|\nabla f(\bar{\boldsymbol{x}}^{(t)})\right\|_{\mathrm{F}}^2\right] \\
\leq &\frac{8}{\gamma T}\left(f(\bar{\boldsymbol{x}}^{(0)}) - f^*\right) + \frac{4\gamma L}{n}\sigma^2 \\
&+ \frac{1}{T}\left(\frac{64L^2}{(1-\beta)^2 n}\left\|\boldsymbol{\Pi}\boldsymbol{X}^{(0)}\right\|_{\mathrm{F}}^2 + \frac{1024\gamma^2 L^2}{(1-\beta)^4 n}\left\|\nabla\boldsymbol{F}(\boldsymbol{X}^{(0)})\right\|_{\mathrm{F}}^2\right) \\
&+ \frac{2048\gamma^2 L^2}{(1-\beta)^4 T}\sigma^2 + \frac{3072\gamma^4 L^4}{(1-\beta)^4 n}\sigma^2 + \frac{512\gamma^2 L^2}{(1-\beta)^3}\sigma^2 \\
\leq &\frac{8}{\gamma T}\left(f(\bar{\boldsymbol{x}}^{(0)}) - f^*\right) + \frac{5\gamma L}{n}\sigma^2 + \frac{2560\gamma^2 L^2}{(1-\beta)^3}\sigma^2 \\
&+ \frac{1}{nT}\left(\frac{64L^2}{(1-\beta)^2}\left\|\boldsymbol{\Pi}\boldsymbol{X}^{(0)}\right\|_{\mathrm{F}}^2 + \left\|\nabla\boldsymbol{F}(\boldsymbol{X}^{(0)})\right\|_{\mathrm{F}}^2\right).
\end{aligned}
\tag{26}
$$

To meet the conditions of Lemma 12, Lemma 13, (25), (26), it suffices to let $\gamma \leq \frac{(1-\beta)^2}{50L}$. Then, (24) follows by setting $g(\gamma)$ to be the RHS of (26), $A = 8\left(f(\bar{\boldsymbol{x}}^{(0)}) - f^*\right)$, $B = \frac{5L}{n}\sigma^2$, $C = \frac{2560L^2}{(1-\beta)^3}\sigma^2$ and $\alpha = \frac{(1-\beta)^2}{50L}$ in Lemma 14. $\qquad\square$

# D    Numerical Experiments

## D.1    Network-size independent consensus rate

In this experiment, we set $M = 5\ln(n)$ for D-EquiStatic and $M = 2\ln(n)$ for U-EquiStatic, which is consistent with Theorems 1 and 3. For OD-EquiDyn and OU-EquiDyn, we set $M = 5\ln(n)$ and $\eta = 0.5$. Fig. 9 shows that the consensus rate is independent of the network size for all EquiTopo graphs. The results are obtained by averaging over 3 independent random experiments.

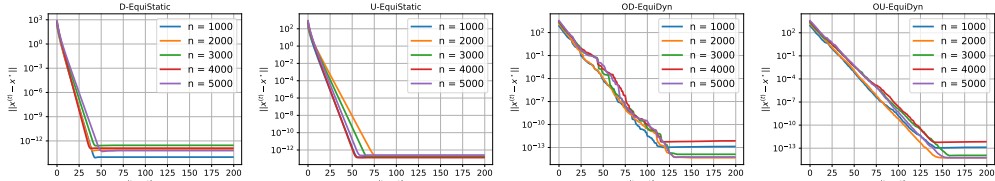

Figure 9: The EquiTopo graphs can achieve network-size independent consensus rates.

## D.2    Comparison with other topologies

We compare the consensus rate between topologies with one-peer or $\Theta(\ln(n))$ neighbors on network-size $n = 300$ and $n = 4900$. In the one-peer case, each topology has exactly one neighbor. For OD-EquiDyn and OU-EquiDyn, we set $M = n-1$ and $\eta = 0.5$. In the $\Theta(\ln(n))$ neighbors case, we set $M = 9$ and $M = 13$ for $n = 300$ and $n = 4900$, respectively, so that the number of neighbors is identical to the static exponential graph for a fair comparison. The results are obtained by averaging over 10 and 3 independent random experiments for $n = 300$ and $n = 4900$, respectively.

## D.3    DSGD with EquiTopo

**Least-square** The distributed least square problems are defined with $f_i(\boldsymbol{x}) = \|\boldsymbol{A}_i\boldsymbol{x} - \boldsymbol{b}_i\|^2$, in which $\boldsymbol{x} \in \mathbb{R}^d$ and $\boldsymbol{A}_i \in \mathbb{R}^{K\times d}$. In the simulation, we let $d = 10$ and $K = 50$. At node $i$, we generate each element in $\boldsymbol{A}_i$ following standard normal distribution. Measurement $\boldsymbol{b}_i$ is generated

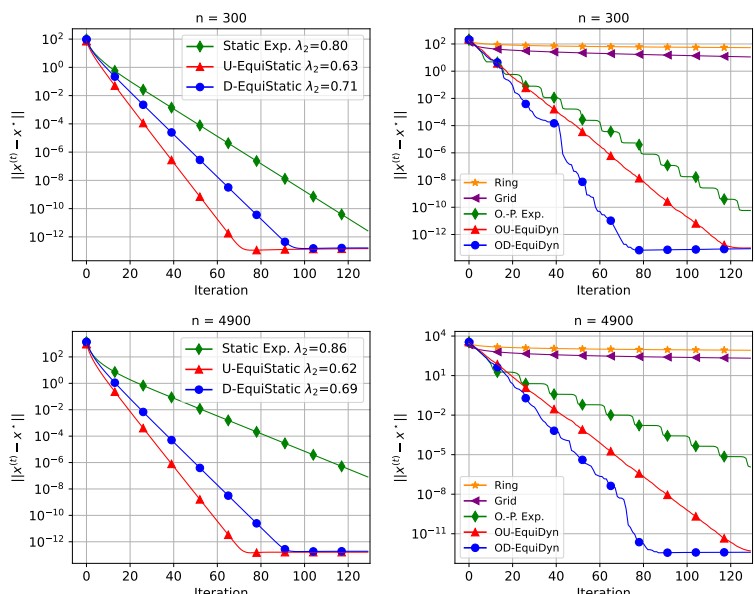

Figure 10: Consensus rate comparison among different network topologies in average consensus problem. Left: all graphs are with $\Theta(\ln(n))$ degree. Right: all graphs are with $\Theta(1)$ degree.

by $\boldsymbol{b}_i = \boldsymbol{A}_i \boldsymbol{x}^\star + \boldsymbol{s}_i$ with a given arbitrary $\boldsymbol{x}^\star \in \mathbb{R}^d$ where $\boldsymbol{s}_i \sim \mathcal{N}(0, \sigma_s^2 \boldsymbol{I})$ is some white noise. At each iteration $t$, each node will generate a stochastic gradient via $\widehat{\nabla f}_i(\boldsymbol{x}) = \nabla f_i(\boldsymbol{x}) + \boldsymbol{n}_i$ where $\boldsymbol{n}_i \sim \mathcal{N}(0, \sigma_n^2 \boldsymbol{I})$ is a white gradient noise. By adjusting constant $\sigma_n$, we can control the noise variance. In this experiment, we set $\sigma_s = 0.1$ and $\sigma_n = 1$. The network size $n$ is 300, and we set $M = 9$ so that D/U-EquiStatic has the same degree as the static exponential graph. After fixing $M$, we sample the basis until the second-largest eigenvalue of the gossip matrix is small enough. The initial learning rate is $0.037$ and decays by $1.4$ every $40$ iterations. The results are obtained by averaging over 10 independent random experiments.

**Deep learning**

**MNIST.** We utilize EquiTopo graphs in DSGD to solve the image classification task with CNN over MNIST dataset [15]. Like the CIFAR-10 experiment, we utilize BlueFog [38] to support decentralized communication and topology settings in a cluster of 17 Tesla P100 GPUs. The network architecture is defined by a two-layer convolutional neural network with kernel size 5 followed by two feed-forward layers. Each convolutional layer contains a max pooling layer and a Rectified Linear Unit (ReLu). We generate D/U-EquiStaic with $M = 4$ and sample OD/OU-EquiDyn with $M = 16$ and $\eta = 0.53$. The local batch size is $64$, momentum is $0.5$, the learning rate is $0.01$, and we train for 20 epochs. Centralized SGD and Ring are included for comparison. See Fig. 11 for the training loss and test accuracy of D/U-EquiStaic and OD/OU-EquiDyn graphs. See Table 5 for the test accuracy calculated by averaging over last 3 epochs. EquiTopo graphs achieve competitive train loss and test accuracy to centralized SGD.

**CIFAR-10.** We use the ResNet-20 model implemented by [10]. In this experiment, we train for 130 epochs with local batch size 8, momentum 0.9, weight decay $10^{-4}$, and the initial learning rate $0.01$, which is divided by 10 at 50th, 100th, and 120th epochs. We follow the data augmentation from [14], a $4 \times 4$ padding followed by a random horizontal flip and a $32 \times 32$ random crop. We generate D/U-EquiStaic with $M = 5$ and sample OD/OU-EquiDyn with $M = 16$ and $\eta = 0.53$. See Fig. 12 for the training loss and test accuracy of OD/OU-EquiDyn. See Table 5 for the test accuracy calculated by averaging over last 5 epochs.

### D.4 DSGT with EquiTopo

In addition to the DSGD experiments, we apply the OD/OU-EquiDyn graphs to the DSGT algorithm when solving logistic regression with non-convex regularizations, i.e., $f_i(\boldsymbol{x}) = \frac{1}{L} \sum_{\ell=1}^{L} \ln(1 +$

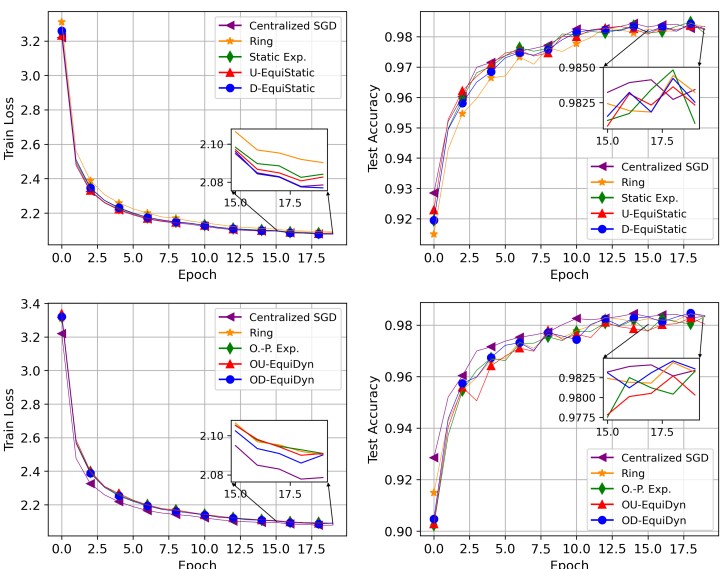

Figure 11: Train loss and test accuracy comparisons among different topologies for CNN on MNIST.

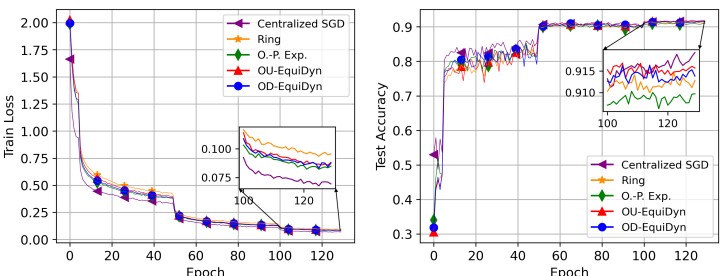

Figure 12: Train loss and test accuracy comparisons among different topologies for ResNet-20 on CIFAR-10.

Table 5: Comparison of test accuracy(%) with different topologies over MNIST and CIFAR-10 datasets.

| Topology | MNIST Acc. | CIFAR-10 Acc. |
|---|---|---|
| Centralized SGD | 98.34 | 91.76 |
| Ring | 98.32 | 91.25 |
| Static Exp. | 98.31 | 91.48 |
| O.-P. Exp. | 98.17 | 90.86 |
| D-EquiStatic | 98.29 | **92.01** |
| U-EquiStatic | 98.26 | 91.74 |
| OD-EquiDyn | **98.39** | 91.44 |
| OU-EquiDyn | 98.12 | 91.56 |

$\exp(-y_{i,\ell} \boldsymbol{h}_{i,\ell}^T \boldsymbol{x})) + R \sum_{j=1}^{d} x_{[j]}^2 / (1 + x_{[j]}^2)$ where $x_{[j]}$ is the $j$-th element of $\boldsymbol{x}$, and $\{\boldsymbol{h}_{i,\ell}, y_{i,\ell}\}_{\ell=1}^{L}$ is the data kept by node $i$. Data heterogeneity exists when local data $\xi_i$ follows different distributions $\mathcal{D}_i$ in problem (7). To control data heterogeneity across the nodes, we first let each node $i$ be associated with a local solution $\boldsymbol{x}_i^\star$, and such $\boldsymbol{x}_i^\star$ is generated by $\boldsymbol{x}_i^\star = \boldsymbol{x}^\star + \boldsymbol{v}_i$ where $\boldsymbol{x}^\star \sim \mathcal{N}(0, \boldsymbol{I}_d)$ is a randomly generated vector while $\boldsymbol{v}_i \sim \mathcal{N}(0, \sigma_h^2 \boldsymbol{I}_d)$ controls the similarity between each local solution. Generally speaking, a large $\sigma_h^2$ results in local solutions $\{\boldsymbol{x}_i^\star\}$ that are vastly different from each other. With $\boldsymbol{x}_i^\star$ at hand, we can generate local data that follows distinct distributions. At node $i$, we generate each feature vector $\boldsymbol{h}_{i,\ell} \sim \mathcal{N}(0, \boldsymbol{I}_d)$. To produce the corresponding label $y_{i,\ell}$, we generate a random variable $z_{i,\ell} \sim \mathcal{U}(0,1)$. If $z_{i,\ell} \le 1 + \exp(-y_{i,\ell} \boldsymbol{h}_{i,\ell}^T \boldsymbol{x}_i^\star)$, we set $y_{i,\ell} = 1$; otherwise $y_{i,\ell} = -1$. Clearly, solution $\boldsymbol{x}_i^\star$ controls the distribution of the labels. This way, we can

easily control data heterogeneity by adjusting $\sigma_h^2$. Furthermore, to easily control the influence of gradient noise, we will achieve the stochastic gradient by imposing a Gaussian noise to the real gradient, i.e., $\widehat{\nabla} f_i(\boldsymbol{x}) = \nabla f_i(\boldsymbol{x}) + \boldsymbol{s}_i$ in which $\boldsymbol{s}_i \sim \mathcal{N}(0, \sigma_n^2 \boldsymbol{I}_d)$. We can control the magnitude of the gradient noise by adjusting $\sigma_n^2$.

We let $d = 10$, $L = 1000$, $n = 300$, $R = 0.001$, and $\sigma_h = 0.2$ in the simulation. For OD/OU-EquiDyn, we set $M = n - 1$ and $\eta = 0.5$. The learning rate for OD/OU-EquiDyn and O.-P. Exp. is 3 and 1.5, respectively so that all of them converge to the same level of accuracy. The left plot in Fig. 13 depicts the performance of different one-peer graphs in DSGT. The right plot in Fig. 13 illustrates how O.-P. Exp. behaves if it has the same learning rate 3 as OU/OD-EquiDyn. The gradient norm is used as a metric to gauge the convergence performance. The results are calculated by averaging over 10 independent random experiments. It is observed that OD/OU-EquiDyn converges faster than a one-peer exponential graph.

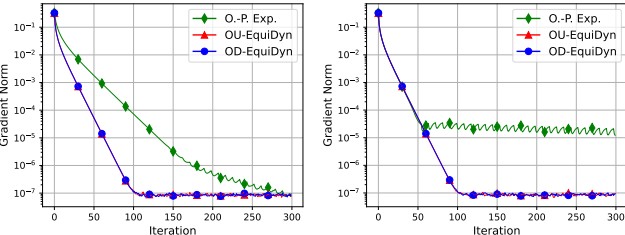

Figure 13: OD/OU-EquiDyn in DSGT. Left: The learning rates for O.-P. Exp. and OU/OD-EquiDyn are 1.6 and 3, respectively, so that all algorithms achieve the same accuracy. Right: The learning rates for all algorithms are 3 so that they share the same convergence rate in the initial stage.