# OpenReview forum: "Communication-Efficient Topologies for Decentralized Learning with $O(1)$ Consensus Rate"
_NeurIPS.cc/2022/Conference — NeurIPS 2022 Accept_

### Official Review · Reviewer_7t7T · 2022-07-01

**Rating:** 6
**Confidence:** 3
**Soundness:** 3 good
**Presentation:** 3 good
**Contribution:** 2 fair

**Summary:**

This work  introduces new families of graph topologies having (almost) constant degree and network size independent mixing rates. This has the advantage that each node needs to exchange information with relatively few neighbors while achieving fast mixing rates. The authors propose directed and undirected version of static graphs with desired properties and then propose dynamic graphs for which the degree of each agent is always 1. As expected, the price to pay for this is a degradation in the mixing rate. Finally, the authors incorporate the families of graph topologies into distributed optimization schemes and analyze the graphs impact on the algorithms' performances.

**Questions:**

I would like the authors to provide some clarification on the topic discussed on the first paragraph in ``Strengths and Weaknesses.'' above.

**Limitations:**

see above

**Strengths And Weaknesses:**

The main weakness lies in the comparison with  the Erd\H{o}s-R'{e}nyi (ER) random graph. While in Table 1 it is stated that ER graphs yield a consensus rate of $1-\Theta(1/\ln(n)),$ this result seems to have been  improved in the literature.  See for instance Proposition 5 in

```Network Topology and Communication–Computation Tradeoffs in Decentralized Optimization'' by A. Nedich, et. al., in IEEE Proceedings 2018,

where it is established that if $p = (1+a)\ln (n)$ one achieves a solution $\varepsilon$ away from consensus in time $\mathcal{O}(\log(\varepsilon^{-1}))$, implying that the rate is invariant with respect to the network size (this is done for undirected graphs). Note that ER random graphs have the additional advantage of guaranteeing that the consensus time will scale as stated above, with a probability $p$ that goes to $1$ as $n \to \infty$. This is in contrast with the families D-EquiStatic and U-EquiStatic  which can be generated algorithmically using Algorithm 3 in Appendix A with the understanding that one can generate a graph with the desired properties with probability $p,$ where $p$ does not improve (go to 1)   as $n$ increases.

On the other hand, the main strength of the paper lies in the use of the structure of the families D-EquiStatic and U-EquiStatic to build the classes OD-EquiDyn and OU-EquiDyn wherein  all agents are connected via a time-varying graph whose each instance has  degree 1. This naturally leads to a loss in the consensus rate, which is still related to the consensus rate of the ``base''  graph of the D-EquiStatic or U-EquiStatic family.

That said, given the discussion in the previous paragraph, I would request the authors to highlight the shortcomings of having each agent randomize transmissions to one of its neighbors over a ``base'' ER graph.

---

> ### Author Response · Authors · 2022-08-02
> **Thanks for your review, we have addressed all your concerns (Part 2)**
>
> **3. Can E.-R. graph admit one-peer variant?**
>
> The reviewer raises an interesting question: *what happens if each node randomizes transmissions to one of its neighbors over a base E.-R. graph?* Our thoughts on this question are as follows.
>
>
> - **No results on one-peer E.-R. graph appear in literature** to the best of our knowledge. Can E.-R. graph admit a one-peer variant? Can the one-peer variant be directed? Is it endowed with a network-size independent consensus rate? All these are **open-questions**. In contrast, the proposed OD/OU-EquiDyn are one-peer graphs with a theoretical $O(1)$ consensus rate and can be applied to both directed and undirected scenarios.
>
> - **It is non-trivial to generate a one-peer E.-R. graph.** According to the reviewer, if each node randomizes transmissions to one of its neighbors over a base E.-R. graph, the resulting topology, with high probability, is **not one-peer**. It is possible that two nodes, when randomly selecting their neighbors, will send messages to the same neighbor and hence break the one-neighbor rule. It seems the generation of one-peer E.-R. also requires some delicate scheduling rules as in OU-EquiDyn.
>
> - There is **no clue** that one-peer E.-R. graph, even if it exsits (probably does not), can converge faster than OD/OU-EquiDyn. It is because the base E.-R. graph is no better than EquiStatic in terms of degree, consensus rate, and generation probability; see discussion in Point 1.
>
>
>
> \
> In summary, while trying very hard, we did not figure out how to generate the one-peer variant based on the E.-R. graph. It does not appear in any existing literature, has no theoretical guarantees to converge, and seems non-trivial to develop. We hope these discussions can resolve the reviewer's comments on the one-peer E.-R. graph.

---

> > ### Comment · Reviewer_7t7T · 2022-08-09
> > **Reply to the Rebuttal**
> >
> > I appreciate the time the authors spent to revise the manuscript and reply to my questions/concerns.
> > My detailed reply follows.
> >
> > *About comment 1:* Thank you for addressing my concerns. I agree with the authors that a suitable choice of $p$ yields equivalent performance of U/D-Equistatic and an E.R. graph. From the authors' response it seems to be that the main advantage of the proposed topologies when compared to E.R. graphs is a strict control on the graphs' maximum degree. I possible, to try to find some result in the literature regarding the probability distribution of the maximum degree of the graph for an E.R. graph. I feel some discussion comparing the proposed topologies and E.R. graphs should be included in the final version of the paper.
> >
> > *About  Comment 2:*  I agree with the authors' reply.   I find the loss in rate natural and expected but appreciate the clarification.
> >
> > *About comment 3:*    I understand no result appears in the literature. However, due to the fact that it is known that the E.R. graphs behave well (by well I mean fulfilling the properties discussed by the authors in the reply to comment 1), the natural question is   whether one may exploit them following a similar strategy to EquiDyn.
> >
> > *Conclusions:* I am overall satisfied with the authors' response, and find the contribution sufficiently novel and somewhat relevant. I will raise my score to 6.

---

> > > ### Author Response · Authors · 2022-08-09
> > > **Many thanks for raising the score. Much appreciated.**
> > >
> > > \
> > > We thank the reviewer for all the valuable comments, helpful discussions, and constructive suggestions during the rebuttal. We will add a detailed comparison with ER graphs in the paper.
> > >
> > > \
> > > We also spent significant time examining the possible ways to generate a series of one-peer graphs from a given E.R. graph. After some preliminary trials, we find it is challenging to develop one-peer ER graphs effectively due to the following facts.
> > >
> > >
> > > \
> > > U/D-EquiStatic graphs are random, but they have a very useful **equidistance** structure that plays an important role to develop one-peer graphs and establish theoretical guarantees. In contrast, ER is a purely random graph in which, while trying very hard, we have not identified a useful structure to facilitate the one-peer extension yet. Our experience in utilizing equidistance structure to develop one-peer graphs in U/D-EquiStatic may not be naturally extended to ER.
> > >
> > > \
> > > On the other hand, in the case of time-varying random graphs, making the algorithm practical could be another concern. In implementations, each node needs to have a new neighbor per iteration. If their neighbor is chosen randomly, some computation/memory overhead is required so that no conflcts (i.e., two or more nodes pick the same neighbor) will appear in the entire network topology. Our algorithm 4 takes advantage of the equidistance structure, and each node only needs O(1) cost to calculate its neighbor. This makes OD/OU-EquiDyn applicable in real-world distributed settings. A naive way of getting one-peer graphs from an E.R. graph may suffer at worst $O(n^2)$ computation and/or memory overhead at each iteration.
> > >
> > > \
> > > We are still thinking about a smart way to generate one-peer ER graph, and will keep the reviewers updated once we figure it out. Thanks again for bringing up this very interesting question!

---

> ### Author Response · Authors · 2022-08-02
> **Thanks for your review, we have addressed all your concerns (Part 1)**
>
> We thank the reviewer for the valuable comments. We have attempted to address them as best as we can. We are glad to clarify any further comments or questions.
>
> **1. Comparison with Erdos-Renyi Graph**
>
> Many thanks for providing the updated results on Erdos-Renyi (E.-R.) Graph. After carefully studying the referred literature, we agree that E.-R. graph can achieve a network-size independent consensus rate in high probability while maintaining an averaged degree of $O(\ln(n))$. We will correct Table 1 accordingly.
>
> Given this newly updated result, we would like to make several clarifications.
>
> - **U-EquiStatic is no worse than E.-R. graph in degree and consensus rate.** The degree and consensus rate of U-EquiStatic are on the **same** order as that of E.-R. graph.
>
> - **The generation probability of EquiStatic is no worse than E.-R. graph.** The reviewer has made an **incorrect** claim that "the generation probability $1 - p$ of U/D-EquiStatic cannot go to $1$ as $n \to \infty$". In lines 452-453 of the supplementary material, it is established that "For any network-size independent consensus rate $\rho \in (0,1)$ and probability $p \in (0,1)$, if $M \ge \frac{8}{3\rho^2}\ln(\frac{2n}{p})$, the  U/D-EquiStatic with consensus rate $\rho$ can be generated with probability $1-p$." If we choose $p=\frac{1}{n}$, U/D-EquiStatic will be generated with probability $1-\frac{1}{n}$ while maintaining degree $M = \frac{16}{3\rho^2}\ln(\sqrt{2}n) = O(\ln(n))$. In this scenario, **the probability to generate U/D-EquiStatic (i.e., $1-\frac{1}{n}$) also goes to $1$ as $n \to \infty,$** which is no worse than E.-R. graph.
>
> - **E.-R. graph is highly unbalanced.** While E.-R. graph achieves $O(\ln(n))$ degree in expectation, it can be highly unbalanced. In other words, some nodes may have much more neighbors than others in a random realization of the E.-R. graph. It is worth noting that the communication overhead in a network topology is **determined by the maximum degree**. Since the maximum degree in E.-R. graph can be $O(n)$ with a non-zero probability, it becomes very slow in some realizations. In contrast, U/D-EquiStatic is guaranteed to have $O(\ln(n))$ maximum degree in theory, which is a highly balanced network that should be more suitable for efficient communication.
>
> - **E.-R. graph is undirected.** Bidirectional communication can be more expensive than unidirectional communication when the channel has limited bandwidth, the transmitted message is of a huge dimension, or the full-duplex communication is not available. In contrast, EquiTopo provides both directed and undirected graphs to fit different applications.
>
> - **It is unclear in the literature whether E.-R. graph can admit one-peer extensions.** However, EquiTopo admits OD/OU-EquiDyn, which has a network-size independent consensus rate while maintaining maximum degree as $1$.
>
>
> \
> In summary, we believe **EquiTopo graphs are important complements to E.-R. graphs**, especially in scenarios where a balanced, directed, or even one-peer graph construction is preferred. One such scenario lies in the data-center GPU cluster; see our response to Reviewer 2. In the revision, we will put all discussions above in the main paper to clarify the comparison with E.-R. graphs.
>
> \
> **2. Does the one-peer variant lead to a loss?**
>
> Yes and no. As the reviewer correctly points out, the one-peer variant can lead to a loss in consensus rate compared to its base EquiStatic graph. However, the rate of the one-peer variant is also on the order of $O(1)$, which is still very fast.
>
> On the other hand, since the per-iteration communication overhead in OD/OU-EquiDyn is $O(1/\ln(n))$ more efficient than EquiStatic, they can save significant wall-clock time in scenarios such as large-scale deep neural network training where the communication overhead is typically the bottleneck. For this reason, the one-peer time-varying graph is extensively studied and used in industry; see references [37, R1, R2, R3].
>
> \
> References (for Part 1):
>
> [R1] Kong et.al., "Consensus Control for Decentralized Deep Learning", ICML 2021
>
> [R2] BlueFog, Github repo at https://github.com/Bluefog-Lib/bluefog
>
> [R3] BAGUA, Github repo at https://github.com/BaguaSys/bagua

---

### Official Review · Reviewer_CH48 · 2022-07-11

**Rating:** 5
**Confidence:** 3
**Soundness:** 3 good
**Presentation:** 3 good
**Contribution:** 3 good

**Summary:**

The paper proposes a new family of topologies which has a constant degree and a network-size-independent consensus rate and can be used to accelerate decentralized optimization and learning algorithms.

**Questions:**

I expect in real applications, the communication topologies are given or constrained.  Does the topology design require each pair of nodes are allowed to communicate?  Is there any examples in which such a topology design is feasible?

**Strengths And Weaknesses:**

The paper has a very novel idea. The paper is well written. The results are very interesting.

The only concern is whether it is feasible to design topologies in this way in real world applications.

---

> ### Author Response · Authors · 2022-08-02
> **Thanks for your review, we have addressed all your concerns**
>
> We thank the reviewer for bringing up this interesting discussion. Our thoughts are summarized as follows.
>
> **1. Scenarios with constrained topology**
>
> We agree with the reviewer that the topology between nodes can be given or constrained in many real applications. Typical examples include wireless sensor networks and Federated Learning, where each node can only communicate with neighbors physically close to it. In these applications, the topology is fixed and cannot be adjusted freely, and we can not apply the proposed EquiTopo graphs.
>
> **2. GPU clusters in data-centers**
>
> But there do exist scenarios in which the topology can be configured freely. One of the most important scenarios is the GPU clusters maintained in data centers, which train large-scale deep learning tasks such as GPT [R1] or DALL-E [R2].
>
> In data-center clusters, all GPUs are physically close to each other, and they are typically maintained in the same room. Moreover, each pair is connected with stable channels such as fibers, InfiniBand, PCI-E, and NVLink. With this setup, all GPUs in a data center can be regarded to be connected by a fully connected network.
>
> [R1] OpenAI GPT-3 https://openai.com/blog/gpt-3-apps/
>
> [R2] OpenAI DALLE-2 https://openai.com/dall-e-2/
>
> **3. Topology design in GPU clusters**
>
> Given a fully connected topology, why don't we use the global averaging? Why do we need to develop various sparse topologies to conduct gossip averaging? The answer is communication efficiency.
>
> The global averaging requires a node to collect messages from the entire network, causing the communication overhead as $O(n)$. This is unbearable in deep learning, where each message is of huge dimension. In contrast, communication over a sparse topology, such as the proposed OD-EquiDyn, can reduce the communication overhead from $O(n)$ to $O(1)$. For this reason, OD-EquiDyn scales well for an arbitrarily large $n$ while global averaging does not.
>
> Topology design is a critical and long-standing problem in decentralized optimization. Nedich et. al. have conducted a detailed study on it in [21]. In the large-scale deep learning era, more sparse and effective topologies show strong empirical performance [R3, 36]. In industry, the decentralized BlueFog system [37, R4] built by researchers in Google and Alibaba utilizes a one-peer exponential graph in GPU clusters to achieve state-of-the-art throughput in deep learning. Another BAGUA system [R5] built by Kwai engineers utilizes the random one-peer graph to achieve similar performances. The success of these two topologies relies on the condition that any two GPUs can be communicated freely. Our work follows this line and comes up with an even better family of EquiTopo graphs.
>
> [R3] Kong et.al., "Consensus Control for Decentralized Deep Learning", ICML 2021
>
> [R4] BlueFog, Github repo at https://github.com/Bluefog-Lib/bluefog
>
> [R5] BAGUA, Github repo at https://github.com/BaguaSys/bagua
>
>
> **4. Experiments in GPU clusters**
>
> Since we narrow the applications of EquiTopo down to deep neural network training, we conducted experiments in Sec. 6 with real GPU clusters. As we have discussed above, the topology in GPU clusters can be configured easily, and we are able to test performances of various topologies such as ring, fully-connected graph, static exponential graph, one-peer exponential graph, D/U-EquiStatic, and OD/OU-EquiDyn. This again illustrates that there exist scenarios in which the topology can be configured freely.
>
>
> **5. A summary**
>
> In summary, the EquiTopo graphs can be applied to scenarios where any pair of nodes can communicate when necessary. An important example is GPU clusters for large-scale deep learning. They cannot be applied to scenarios where the topology is fixed or constrained.
>
>
> **6. Future directions of EquiTopo**
>
> In future works, we will try to extend the idea behind EquiTopo to scenarios with constrained (not fully connected) topologies. Perhaps we can build a sparser (or even one-peer) variant of the given constrained topology while maintaining the same consensus rate.
>
> We hope these discussions can address the reviewer's concerns. We are glad to clarify any further comments or questions.

---

> ### Author Response · Authors · 2022-08-09
> **Can we have your response to our rebuttals**
>
> \
> Dear Reviewer CH48,
>
> \
> Thanks very much for your very positive comments. We are happy that you find our work **very novel** and **very interesting**.
>
> \
> Topology design is one of the **fundamental and long-standing question** in decentralized optimization, and it meets **new settings and challenges** (i.e., communication efficiency) in the large scale deep learning era. We hope our rebuttal has clarified the applications of the proposed topology on large-scale GPU clusters.
>
> \
> Since the author-reviewer discussion period will end very soon. Can we have your response to our rebuttals? We are happy to clarify any further comments or questions.
>
> \
> Best,\
> Authors

---

### Official Review · Reviewer_fyHE · 2022-07-11

**Rating:** 7
**Confidence:** 5
**Soundness:** 3 good
**Presentation:** 3 good
**Contribution:** 3 good

**Summary:**

This article discusses network structure for decentralized consensus and optimization. The authors propose EquiTopo, a family of network structures with $O(1)$ consensus rates, i.e., network-independent rates. For both directed and undirected topologies, the authors present static network structures with degrees $O(\ln n)$ and propose counterpart dynamic (time-varying) networks to relax connection degrees to $O(1)$. Lastly, the authors apply their proposed network topologies to decentralized stochastic gradient descent as well as decentralized stochastic gradient tracking algorithms and provide a comparison of the number of rounds required for convergence.

**Questions:**

- Which of the following terms in the convergence rate describes transient iterations (Trans. Iters.)? This needs to be clarified in the description.
It is often necessary to use push-sum algorithms (gradient-push algorithms) for directed networks (static or dynamic). There are, however, no differences between the algorithms proposed for directed and undirected networks in this study. Does this only apply to the network design (EquiTopo) proposed here? In that case, please elaborate in the manuscript.
- Do the theoretical results for DSGD and DSGT presented (Theorem 5 and Theorem 6) represent new results? If not, it is best to present them as propositions and to properly cite the seminal work.

**Limitations:**

We have limited control over the network design in decentralized optimization, which is the main limitation of this work. The authors of this paper have pointed out that their work is primarily concerned with defining the communication structure on a central machine with multiple GPUs, i.e., high-throughput scenarios with the capability of making arbitrary connections between entities. However, this challenges robustness extensions where entities may fail to communicate synchronously.

**Strengths And Weaknesses:**

Strengths:
- A novel result is presented for network design in decentralized optimization problems where the consensus rate is independent of the size of the network.
- By reducing the connection degree from $O(\ln n)$ to $O(1)$, constant communication can be achieved per iteration.

Weakness:
- As shown in Theorem 5 and Theorem 6, the convergence rate improvement for DSGD and DSGT only occurs in transient terms.

---

> ### Author Response · Authors · 2022-08-02
> **Thanks for your review, we have addressed all your comments (Part 2)**
>
> **3. Push-sum**
>
> The push-sum technique is mainly used when the mixing matrix is only column-stochastic (but not doubly stochastic). However, the EquiTopo graphs we constructed in the paper are always **doubly stochastic**, meaning that both the row and column sum to 1. By designing topologies like this, the push-sum technique is not needed. For example, in the well-known gradient tracking paper [22], Algorithm 1 does not utilize push-sum for doubly-stochastic (but possibly directed) graphs.
>
>
> **4. Theorem 6 has new results**
>
> As clarified in lines 225 -- 227 of the paper, we follow the proof from [12] and apply EquiTopo to achieve Theorem 5. We will rephrase it as a proposition in the revision.
>
> On the other hand, Theorem 6 does have new results. Existing works on DSGT assume the weight matrix to be either symmetric [1, 11] or static [34]. However, the proposed OD/OU-EquiDyn is neither symmetric nor static. Theorem 6 admits a new, improved convergence rate for stochastic decentralized optimization over *asymmetric* or *time-varying* weight matrix.
>
> **5. Topology design in GPU clusters**
>
> While EquiTopo has both efficient communication and a network-size independent consensus rate, we agree with the reviewer that it may suffer from the robustness issue. When one pair of nodes cannot communicate as we schedule, EquiTopo may break down. We will clarify this limitation in the revision.
>
> However, EquiTopo has significant values in data-center GPU clusters, which are used to train large-scale deep learning tasks. In data-center clusters, all GPUs are physically close to each other, and they are typically maintained in the same room. Moreover, each pair is connected with high-performance and stable channels such as fibers, InfiniBand, PCI-E, and NVLink. In some cases, a team of infrastructure engineers may be even hired for cluster maintenance. We can regard the GPU cluster as a **reliable** network in which the topology can be **fully controlled**, which differs from the wireless networks where the communication between two nodes is ad-hoc. As a result, the topology design to enhance communication efficiency is very much appreciated in GPU clusters. A line of research in this direction has been ongoing; see references [2, 21, 33, 36, R3, R6]. In industry, decentralized deep learning systems such as BlueFog [37, R7] and BaGua [R8] utilize delicately-designed network topologies to achieve much higher throughput than traditional centralized systems.
>
> In future work, we will involve robustness as a major consideration in topology design.
>
>
>
> \
> References (Part 2):
>
> [R6] S. Gan et. al. "Scaling up Distributed Learning with System Relaxations", arXiv 2107.01499, 2021.
>
> [R7] BlueFog, Github repo at https://github.com/Bluefog-Lib/bluefog
>
> [R8] BAGUA, Github repo at https://github.com/BaguaSys/bagua

---

> ### Author Response · Authors · 2022-08-02
> **Thanks for your review, we have addressed all your comments (Part 1)**
>
> We thank the reviewer for the valuable comments. We have attempted to address them as best as we can. We are glad to clarify any further comments or questions.
>
> **1. Transient iteration complexity**
>
> In lines 611-616 of the supplementary material, we have explained how to calculate the transient iteration complexity. We repeat it here to address the reviewer's concerns. We take DSGD as an example. DSGD converges at rate
>
>
> $O(\underbrace{\frac{\sigma}{\sqrt{nT}}}\_{\mbox{Linear speedup}} + \underbrace{\frac{\beta^{\frac{2}{3}}\sigma^{\frac{2}{3}}}{T^{\frac{2}{3}}(1-\beta)^{\frac{1}{3}}} + \frac{\beta^{\frac{2}{3}}b^{\frac{2}{3}}}{T^{\frac{2}{3}}(1-\beta)^{\frac{2}{3}}} + \frac{\beta}{T(1-\beta)}}\_{\mbox{Extra overhead}}),$
>
> which is decomposed into a linear speedup term and an extra overhead. The **extra overhead terms** decide the transient iteration complexity of decentralized algorithms. See details below.
>
> - **Linear speedup**. When $T$ is sufficiently large, the first term $O(1/\sqrt{nT})$ dominates the rate. With the linear speedup term, DSGD requires $T =
> O(1/(n\epsilon^2))$ iterations to reach the desired accuracy $\epsilon$, which is inversely proportional to $n$. We say an algorithm is in its linear-speedup stage at  $T$-th iteration if, for this $T$, the term involving $nT$ is dominating the rate.
>
> - **Transient iterations**. Transient iterations are referred to as those iterations before an algorithm reaches its linear-speedup stage, that is, iterations $1, \dots, T$ where $T$ is relatively small, so non-$nT$ terms (i.e., the extra overhead terms in DSGD) still dominate the rate. For example, to achieve linear speedup in DSGD, $T$ has to satisfy $\frac{\sigma}{\sqrt{nT}} \ge \frac{\beta^{\frac{2}{3}}\sigma^{\frac{2}{3}}}{T^{\frac{2}{3}}(1-\beta)^{\frac{1}{3}}}$, $\frac{\sigma}{\sqrt{nT}} \ge \frac{\beta^{\frac{2}{3}}b^{\frac{2}{3}}}{T^{\frac{2}{3}}(1-\beta)^{\frac{2}{3}}}$, and $\frac{\sigma}{\sqrt{nT}} \ge \frac{\beta}{T(1-\beta)}$, respectively, which turns out to be $T \ge \max\\{\frac{\beta^4n^3}{(1-\beta)^2 \sigma^2}, \frac{\beta^4b^4n^3}{(1-\beta)^4\sigma^6}, \frac{\beta^2 n}{(1-\beta)^2 \sigma^2}\\}$. If $b^2$ and $\sigma^2$ are regarded as constants, we have transient iteration complexity as $T = O(\frac{n^3}{(1-\beta)^4})$. In other words, DSGD has to experience $O(\frac{n^3}{(1-\beta)^4})$ iterations before it achieves the linear speedup stage.
>
>
> We thank the reviewer for pointing out the confusion on transient iterations. We will move the related description from Appendix to the main body of the paper in the revision.
>
>
> **2. Improving extra overhead terms only is not a weakness**
>
>
> We respectfully disagree with the reviewer on this point. Our clarification is listed as follows (when the reviewer talks about *transient terms*, we think the reviewer actually indicates the *extra overhead terms* in the above DSGD rate).
>
> - **Linear speedup term cannot be improved.** It is well recognized in the literature that the linear speedup term $O(\sigma/\sqrt{nT})$ cannot be improved by any distributed or decentralized algorithms utilizing general stochastic gradient oracles, see [R1, R2, R5]. In other words, one cannot develop a decentralized algorithm to achieve a faster linear speedup term than $O(\sigma/\sqrt{nT})$ in the general non-convex and smooth setting. A decentralized algorithm's best result is to make the extra overhead as small as possible.
>
> - **Improving extra overhead is meaningful.** While a decentralized algorithm will achieve the linear speedup stage (i.e., a stage where $\sigma/\sqrt{nT}$ dominates) asymptotically, it may experience **massive transient iterations** to reach that stage. An algorithm may never escape from its transient stage if the iteration budget is limited. This motivates us to improve the extra overhead in DSGD and DSGT by developing better network topologies and thus shortening their transient stage to a state-of-the-art order $O(n^3)$ while maintaining $O(1)$ communication overhead per iteration.
>
> - **Improving extra overhead is a critical topic in decentralized optimization**. Many recent advanced decentralized algorithms are targeting to reduce extra overhead terms. For example, to improve the extra overhead terms, [33, 36, R3] have developed effective network topologies, [11, 34, R4] have removed the influence of data heterogeneity, and [R5] have utilized multiple gossip loops in algorithm constructions.
>
>
> \
> References (Part 1):
>
> [R1] G. Lan, "An Optimal Method for Stochastic Composite Optimization", Mathematical Programming, 2012.
>
> [R2] O. Dekel et. al., "Optimal Distributed Online Prediction Using Mini-Batches", JMLR, 2012
>
> [R3] L. Kong et. al., "Consensus Control for Decentralized Deep Learning", ICML 2021
>
> [R4] S. Alghunaim et. al., "A unified and refined convergence analysis for non-convex decentralized learning", TSP 2022
>
> [R5] Y. Lu et. al. "Optimal complexity in decentralized training", ICML 2021.

---

> ### Author Response · Authors · 2022-08-09
> **Can we have your response to our rebuttals?**
>
> \
> Dear Reviewer fyHE,
>
> \
> Thanks very much for your positive comments. We are happy that you find our work novel.
>
> \
> We have addressed your questions in our rebuttal (especially those on transient iteration complexity, push-sum algorithm, and new results provided by Theorem 6). Since the reviewer-author discussion period will end in a couple of hours, can we have your response to our rebuttals? We are happy to address any further questions or comments.
>
> \
> Best,\
> Authors

---

### Author Response · Authors · 2022-08-06
**Dear Reviewers, can we have your response to our rebuttals?**

\
Dear Reviewers,

\
Thanks very much for your careful review and valuable feedback. We are very happy that all of you have acknowledged that our contributions are novel. For example, some comments are


> The paper has a **very novel idea**. The results are **very interesting**. (From reviewer CH48).

> A **novel** result is presented for network design. (From reviewer fyHE).

> The main strength of the paper lies in ... a time-varying graph whose each instance has degree 1. (From reviewer 7t7T).

We really appreciate your encouraging comments.

\
Your main concerns lie in the applicable scenarios of the proposed graph, as well as a comparison with an improved Erdos-Renyi graph result. We have provided a detailed clarification to each of them.

\
Since the author-reviewer discussion period will end soon, can we have your kind response to our rebuttals? If our response have clarified your concern, could you please re-evaluate our work?

\
Many Thanks! \
Authors

---

### Meta-Review · Area_Chair_QJ5a · 2022-08-24

**Recommendation:** Accept
**Confidence:** Less certain

**Metareview:**

This paper introduces a framework for communication scheduling in fully connected networks, in order to improve the convergence rate of decentralized learning algorithms (specifically, the mixing rate of the underlying consensus process). Both broadcast/convergecast (one-to-many, many-to-one) and directed/undirected communication is considered. I believe that this is a useful addition to the toolset for decentralized optimization on fully connected networks and therefore recommend acceptance. A limitation not raised by the reviewers appears to be the lack of an efficient method for constructing the graph, as it appears that Algorithm 3 essentially performs a random search that could take a long time to converge if rho is selected close to its limit (or that would not converge if rho was chosen too aggressively).

**Award:**

No

---

### Decision · Program_Chairs · 2022-09-14

Accept